# MeGraph: Graph Representation Learning on Connected Multi-scale Graphs

## Abstract

We present `MeGraph`, a novel network architecture for graph-structured data. Given any input graph, we create multi-scale graphs using graph pooling. Then, we connect them into a mega graph by bridging inter-graph edges according to the graph pooling results. Instead of universally stacking graph convolutions over the mega graph, we apply general graph convolutions over intra-graph edges, while the convolutions over inter-graph edges follow a bidirectional pathway to deliver the information along the hierarchy for one turn. Graph convolution and graph pooling are two core elementary operations of `MeGraph`. In our implementation, we adopt the graph full network (GFuN) and propose the stridden edge contraction pooling (S-EdgePool) with adjustable pooling ratio, which are extended from conventional graph convolution and edge contraction pooling. The `MeGraph` model enables information exchange across multi-scale graphs, repeatedly, for deeper understanding of wide range correlations in graphs. This distinguishes `MeGraph` from many recent hierarchical graph neural networks like Graph U-Nets. We conduct comprehensive empirical studies on tens of public datasets, in which we observe consistent performance gains comparing to baselines. Specifically, we establish 5 new graph theory benchmark tasks that require long-term inference and deduction to solve, where `MeGraph` demonstrates dominated performance compared with popular graph neural networks.

## 1 Introduction

In real-world applications, many types of data can be naturally organized as graphs, such as social networks, traffic networks and biological data. Recent advances in graph neural networks (GNNs) have inherited the great success of convolutional neural networks (CNNs) from images to deal with graph-structured data. Popular methods include the GCN (Kipf & Welling, 2016), GIN (Xu et al., 2018), GAT (Vaswani et al., 2017) and Graph U-Nets (Gao & Ji, 2019), etc.

Generally, the development of both CNNs and GNNs is co-evolved, and most effective experiences identified in CNNs are also helpful for GNNs. For example, we have witnessed coupled networks for image and graph data, like CNN vs. GCN, attentional CNN vs. GAT (Vaswani et al., 2017), and U-Net (Ronneberger et al., 2015) vs. Graph U-Net (Gao & Ji, 2019), etc.

Instead of directly transferring advances in CNNs to GNNs, we investigate inherent characteristics in graphs and design a new architecture accordingly. We use the following example to motivate the story. Consider the problem of identifying the shortest path in a chain graph. Using normal graph convolutions, we have to stack multiple graph convolutional layers to enlarge the receptive field to cover the source and the destination nodes. However, if the architecture could infer from a larger scope, e.g., constructing multi-scale graphs in a hierarchy, the shortest path is easier to be estimated by aggregating and delivering information from multi-level scopes. In addition, a single turn of information aggregation or delivery over the hierarchical structure might not be sufficient, because estimation should be refined and deduced over and over again to achieve sure conclusions. That is, the architecture has to repeat the information exchange across the hierarchy multiple times to identify the shortest path for sure. This example will be investigated in our experiment in Section 4. In fact, there have been several recent GNNs working on a hierarchical graph structure. The Graph U-Nets (Gao & Ji, 2019) forms a hierarchy by downsampling the graph with iterative convolutions and top-$k$ pooling, and then upsampling the pooled graph with iterative convolutions and unpooling operators. However, the U-shaped net only propagates the information for a single turn. The GraphFPN (Zhao et al., 2021) builds mappings between the image and graph feature pyramids according to the superpixel hierarchy, and it applies GNN layers on the hierarchical graph to exchange

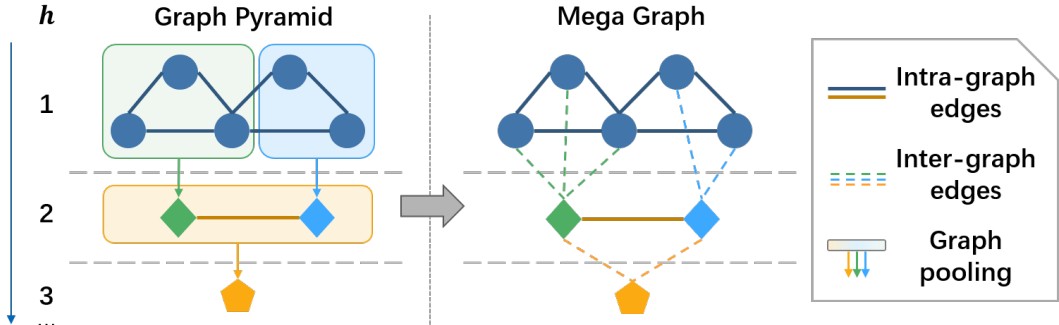

Figure 1: Illustration for comparing the graph pyramid and the mega graph. The graph pyramid is formed with iterative graph pooling. Different shapes represent the nodes in different scales (heights). The inter-graph edges generated during graph pooling connect the graph pyramid into a complete mega graph.

information within the graph pyramid; while the flow of inference still propagates for a single pass over a fixed contextual-hierarchical-contextual structure, as shown in Fig. 1 of (Zhao et al., 2021).

In this paper, we provide a novel perspective for hierarchical graph representation learning. We use differentiable graph pooling methods to create mult-scale graphs, which were also referred to as the graph pyramid in previous methods (Zhao et al., 2021). Conditioning on the graph pooling results, we explicitly connect multi-scale graphs into a mega graph according to how the nodes are pooled together (illustrated in Fig. 1). A straightforward way to learn on the mega graph is to adopt the naive message-passing strategy, which abandons the hierarchical prior knowledge. Instead, we convolve the intra-graph edges and inter-graph edges separately. That is, we stack general graph convolutions over intra-graph edges, while convolutions over inter-graph edges follow a bidirectional pathway to deliver the information along the hierarchy top-down and then reverse back. This process will be repeated multiple times according to two dimensions, i.e., the height of the graph hierarchy and the depth of stacked layers. To realize the above scheme, we adopt two core elementary operations, graph full network (GFuN) and stridden edge contraction pooling (S-EdgePool), which are extended from conventional graph convolution and edge contraction pooling. We conduct comprehensive experiments on tens of public datasets, in which we observe consistent performance gains compared to baselines. Specifically, we establish five new graph theory benchmark datasets that require long-term inference and deduction to solve. In these tasks, `MeGraph` demonstrates dominated performance compared with popular graph neural networks.

Our contributions can be summarized as follows. 1) We propose a novel mega graph structure with general usage for graph neural networks. Given the mega graph, we propose a specific network module to enable repeated information exchange across multi-scale graphs. 2) To control the scale of pooled graphs, we design the S-EdgePool operator, which allows variable pooling stride and pooling ratio. 3) We create five new graph theory benchmark tasks, including problems of shortest path, maximum connected component, graph diameter, etc. The `MeGraph` model achieves obvious improvement on most of the benchmarks compared to popular GNNs.

## 2 NOTATIONS, BACKGROUNDS AND PRELIMINARIES

Let $\mathcal{G} = (\mathcal{V}, \mathcal{E})$ be a graph with node set $\mathcal{V}$ (of cardinality $N^v$) and edge set $\mathcal{E}$ (of cardinality $N^e$). The edge set can be represented as $\mathcal{E} = \{(s_k, t_k)\}_{k=1:N^e}$, where $s_k$ and $t_k$ are the indices of the source and target nodes connected by edge $k$. We define $\mathbf{X}^{\mathcal{G}}$ as features of graph $\mathcal{G}$, which is a combination of global (graph-level) features $\mathbf{u}^{\mathcal{G}}$, node features $\mathbf{V}^{\mathcal{G}}$, and edge features $\mathbf{E}^{\mathcal{G}}$. Accordingly, we use $\mathbf{V}_i^{\mathcal{G}}$ to represent the features of a specific node $v_i$, and $\mathbf{E}_k^{\mathcal{G}}$ denotes the features of a specific edge $(s_k, t_k)$. We may abuse the notations by omitting the superscript $\mathcal{G}$ when there is no ambiguity from the contexts.

### 2.1 GRAPH NETWORK (GN) BLOCK

We follow the graph networks (GN) framework in (Battaglia et al., 2018). Using our notations, a GN block takes a graph $\mathcal{G}$ and features $\mathbf{X} = (\mathbf{u}, \mathbf{V}, \mathbf{E})$ as inputs, and the block outputs new features $\mathbf{X}' = (\mathbf{u}', \mathbf{V}', \mathbf{E}')$. A full GN block (Battaglia et al., 2018) contains the following computational steps (where $\phi$ in each step below indicates an update function that is usually a neural network):

1. Update edge features: $\mathbf{E}_k' = \phi^e(\mathbf{E}_k, \mathbf{V}_{s_k}, \mathbf{V}_{t_k}, \mathbf{u}), \forall k \in [1 \dots N^e]$.

2. Update node features: $\mathbf{V}_i' = \phi^v(\rho^{e \rightarrow v}(\{\mathbf{E}_k'\}_{k \in [1 \dots N^e], t_k = i}), \mathbf{V}_i, \mathbf{u}), \forall i \in [1 \dots N^v]$, where $\rho^{e \rightarrow v}$ is an edge-to-node aggregation function taking the features of incoming edges as inputs.

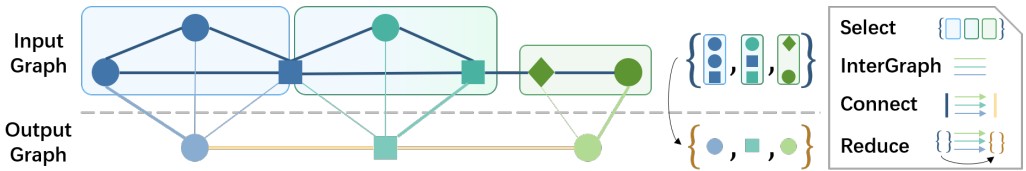

Figure 2: Illustration of graph pooling with SELECT, CONNECT and REDUCE steps. The SELECT function groups subset of nodes in the input graph to form a new node and connect the subset of nodes with the new node via inter-edges. The CONNECT function maps the edges of the input graph to new edges in the pooled graph. The REDUCE function aggregates features of the input graph according to the inter-graph. We only illustrate the reduction of node features for simplicity.

3. Update global features: $\mathbf{u}' = \phi^u(\rho^{e \to u}(\mathbf{E}'), \rho^{v \to u}(\mathbf{V}'), \mathbf{u})$, where $\rho^{e \to u}$ and $\rho^{v \to u}$ are two global aggregation functions over edge and node features.

Given a fixed graph structure $\mathcal{G}$ and the consistent input and output formats defined above, GN blocks can be easily applied to compose deep graph networks. A common *encode-process-decode* architecture design adopted in typical graph networks (Battaglia et al., 2018; Hamrick et al., 2018) is applying the encoding GN block ($\text{GN}_{\text{enc}}$), multiple core GN blocks ($\text{GN}_{\text{core}}$) and the decoding GN block ($\text{GN}_{\text{dec}}$) sequentially on inputs $\mathbf{X}_{\text{input}}$ to obtain the outputs $\mathbf{X}_{\text{output}}$.

## 2.2 GRAPH POOLING

Similar to the concept of pooling in CNNs, graph pooling downsamples the graph structure and reduces the corresponding features, while preserving both structural and semantic graphical information. Following (Grattarola et al., 2022), we define graph pooling as a class of functions POOL that maps a graph $\mathcal{G} = (\mathcal{V}, \mathcal{E})$ with $N^v$ nodes and features $\mathbf{X}^{\mathcal{G}}$ to a reduced graph $\tilde{\mathcal{G}} = (\tilde{\mathcal{V}}, \tilde{\mathcal{E}})$ with $N^{\tilde{v}}$ nodes and new features $\mathbf{X}^{\tilde{\mathcal{G}}}$, where $N^{\tilde{v}} \leq N^v$ and $(\tilde{\mathcal{G}}, \mathbf{X}^{\tilde{\mathcal{G}}}) = \text{POOL}(\mathcal{G}, \mathbf{X}^{\mathcal{G}})$.

The POOL function consists of the following steps SELECT, CONNECT and REDUCE:

$$(\hat{\mathcal{G}}, \mathbf{X}^{\hat{\mathcal{G}}}) = \text{SELECT}(\mathcal{G}, \mathbf{X}^{\mathcal{G}}); \quad \tilde{\mathcal{G}} = \text{CONNECT}(\mathcal{G}, \hat{\mathcal{G}}, \mathbf{X}^{\hat{\mathcal{G}}}); \quad \mathbf{X}^{\tilde{\mathcal{G}}} = \text{REDUCE}(\mathbf{X}^{\mathcal{G}}, \hat{\mathcal{G}}, \mathbf{X}^{\hat{\mathcal{G}}}). \quad (1)$$

The SELECT function maps the nodes in the input graph to the nodes in the pooled graph. Specifically, it creates $N^{\tilde{v}}$ nodes for the pooled graph and connects each node $\tilde{v}$ to a subset of nodes $S_{\tilde{v}} \subseteq \mathcal{V}$ in the input graph. This forms an *undirected* bipartite graph $\hat{\mathcal{G}} = (\hat{\mathcal{V}}, \hat{\mathcal{E}})$, where $\hat{\mathcal{V}} = \mathcal{V} \cup \tilde{\mathcal{V}}$ and $(v, \tilde{v}) \in \hat{\mathcal{E}}$ if and only if $v \in S_{\tilde{v}}$. We call this graph $\hat{\mathcal{G}}$ the inter-graph, which is a larger graph connecting the nodes in the input graph $\mathcal{G}$ and the nodes in the pooled graph $\tilde{\mathcal{G}}$. The SELECT function can generalize to introduce inter-graph features $\hat{\mathbf{X}}^{\hat{\mathcal{G}}}$. For example, we can introduce some edge weights for some edge $(\hat{s}_k, \hat{t}_k)$ in graph $\hat{\mathcal{G}}$ to measure the importance that the node $\hat{s}_k$ from the input graph contributes to the node $\hat{t}_k$ in the pooled graph. The CONNECT function rebuilds the edge set $\tilde{\mathcal{E}}$ between the nodes in $\tilde{\mathcal{V}}$ of the pooled graph $\tilde{\mathcal{G}}$ according to the original edges in $\mathcal{E}$ and the inter-graph edges in $\hat{\mathcal{E}}$. The REDUCE function computes the graph features $\mathbf{X}^{\tilde{\mathcal{G}}}$ of graph $\tilde{\mathcal{G}}$ by aggregating input graph features $\mathbf{X}^{\mathcal{G}}$ according to both the inter-graph $\hat{\mathcal{G}}$ and features $\mathbf{X}^{\hat{\mathcal{G}}}$.

In contrast to the REDUCE function, we further define the EXPAND function in the reversed direction: $\mathbf{X}^{\mathcal{G}} = \text{EXPAND}(\mathbf{X}^{\tilde{\mathcal{G}}}, \hat{\mathcal{G}}, \mathbf{X}^{\hat{\mathcal{G}}})$. Note that the inter-graph $\hat{\mathcal{G}}$ and features $\mathbf{X}^{\hat{\mathcal{G}}}$ can be reused when applying REDUCE or EXPAND to any features of graph $\mathcal{G}$ or $\tilde{\mathcal{G}}$.

By extending the general SELECT-REDUCE-CONNECT framework (Grattarola et al., 2022), our formulation of the POOL function covers most of the current graph pooling methods, including the recent node clustering pooling and node drop pooling methods (Liu et al., 2022). For example, we will explain EdgePool (Diehl et al., 2019) under the scope of this formulation in Section 3.3.

## 3 METHODS

The main idea in our approach is that we explicitly connect multi-scale graphs into a mega graph. We then apply graph neural networks over the mega graph to enable repeated information exchange across multi-scale graphs. In this section, we first introduce how we obtain and connect the multi-scale graphs using graph pooling methods (Section 3.1). Then, we introduce the MeGraph model, which learns hierarchical graph representation on the mega graph through repeated cross-scale information exchange (Section 3.2), followed by specific choices of core modules and innovations made therein (Section 3.3). At last, we analyse the computational complexity of MeGraph (Section 3.4).

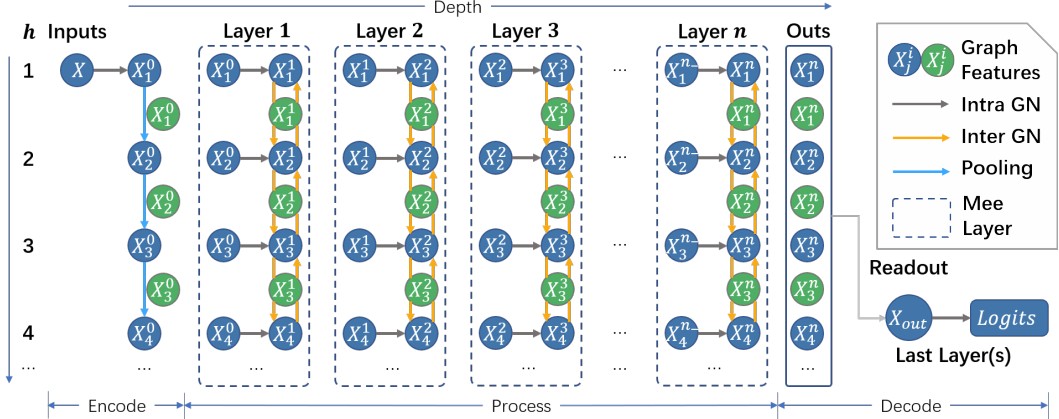

Figure 3: Illustration of the `MeGraph` model where $n_-$ means $n-1$. The blue and green circles represent features of intra- and inter-graphs, respectively. The mega graph is built using graph pooling during the *encode* stage. The `Mee` layer with bidirectional pathways crossing multiple scales is stacked $n$ times during the *process* stage. There are residual links within the $i$-th `Mee` layer for both intra- and inter-graph features from $\mathbf{X}_j^{i-1}$ to $\mathbf{X}_j^i$ (blue ones) and from $\hat{\mathbf{X}}_j^{i-1}$ to $\hat{\mathbf{X}}_j^i$ (green ones) for all height $j$. The multi-scale features are read out during the *decode* stage. The golden inter GN blocks forms bidirectional pathways across multi-scale features.

## 3.1 CONNECTING MULTI-SCALE GRAPHS INTO A MEGA GRAPH

Similar to the image pyramid (Adelson et al., 1984), a graph pyramid is piled up by multi-scale graphs obtained through iteratively downsampling the smallest one using graph pooling. Formally, according to the image feature pyramid (Lin et al., 2017), we define a graph feature pyramid as a set of graphs $\boldsymbol{\mathcal{G}}_{1:h} := \{\mathcal{G}_i\}_{i=1,\cdots,h}$ and features $\mathbf{X}^{\boldsymbol{\mathcal{G}}_{1:h}} := \{\mathbf{X}^{\mathcal{G}_i}\}_{i=1,\cdots,h}$, where $\mathcal{G}_1$ indicates the original graph, $\mathbf{X}^{\mathcal{G}_1}$ denotes the initial features, $h$ is the height of the graph feature pyramid and $(\mathcal{G}_i, \mathbf{X}^{\mathcal{G}_i}) = \texttt{POOL}(\mathcal{G}_{i-1}, \mathbf{X}^{\mathcal{G}_{i-1}})$ for $i > 1$.

By iteratively applying the `POOL` function, we collect the inter-graphs $\hat{\boldsymbol{\mathcal{G}}}_{1:h} := \{\hat{\mathcal{G}}_i\}_{i=1,\cdots,h-1}$ and features $\mathbf{X}^{\hat{\boldsymbol{\mathcal{G}}}_{1:h}} := \{\mathbf{X}^{\hat{\mathcal{G}}_i}\}_{i=1,\cdots,h-1}$ (since the highest inter-graph for height $h$ is $\mathcal{G}_{h-1}$ instead of $\mathcal{G}_h$), where $(\hat{\mathcal{G}}_i, \mathbf{X}^{\hat{\mathcal{G}}_i}) = \texttt{SELECT}(\mathcal{G}_i, \mathbf{X}^{\mathcal{G}_i})$ for $i < h$. Recalling that `SELECT` is the first step of the `POOL` function, the bipartite inter-graph $\hat{\mathcal{G}}$ and features $\mathbf{X}^{\hat{\mathcal{G}}}$ essentially reveals the relationships between the graphs before and after pooling (see also Section 2.2).

Finally, we wire the graph pyramid $\boldsymbol{\mathcal{G}}_{1:h}$ using the edges in the bipartite graphs $\hat{\boldsymbol{\mathcal{G}}}_{1:h}$, resulting in a mega graph $\mathcal{MG} = (\mathcal{MV}, \mathcal{ME})$, where $\mathcal{MV} = \bigcup_{i=1}^h \mathcal{V}_i$ and $\mathcal{ME} = \bigcup_{i=1}^h \mathcal{E}_i \cup \bigcup_{i=1}^{h-1} \hat{\mathcal{E}}_i$. We denote $\mathcal{MG}_{\text{intra}} = \bigcup_{i=1}^h \mathcal{G}_i$ as the intra-graph of $\mathcal{MG}$ and name the edges therein as the intra-edges. Accordingly, $\mathcal{MG}_{\text{inter}} = \bigcup_{i=1}^{h-1} \hat{\mathcal{G}}_i$ is referred to as the inter-graph of $\mathcal{MG}$ and the corresponding edges are called inter-edges. The features $\mathbf{X}^{\mathcal{MG}}$ of mega graph $\mathcal{MG}$ is a combination of intra-graph features $\mathbf{X}^{\boldsymbol{\mathcal{G}}_{1:h}}$ and inter-graph features $\mathbf{X}^{\hat{\boldsymbol{\mathcal{G}}}_{1:h}}$.

## 3.2 THE MEGRAPH MODEL

The most straightforward way to use the mega graph $\mathcal{MG}$ is treating it as an ordinary graph and universally applying graph neural networks like GCNs (Kipf & Welling, 2016) on it. However, our intuition in proposing the mega graph structure is to facilitate the information exchange across multi-scale graphs, while the above method suffers from long-distance message propagation by ignoring the inherent structure of the mega graph. For example, by applying normal GCNs over the mega graph, the features of the original graph $\mathcal{G}_1$ have to be convolved at least $h-1$ times to reach the features of the smallest graph $\mathcal{G}_h$, and vice versa. This is similar to stacking CNNs on a large image, where the pixels at the upper-left corner and the bottom-right corner are receptive in one kernel only at very deep layers.

To overcome this, we propose the `MeGraph` network. The overall architecture of `MeGraph` is illustrated in Fig. 3. As we can observe, the `MeGraph` network follows the common *encode-process-decode* architecture design and uses GNs (see Section 2.1) as elementary building blocks. In the *encode* stage, the feature embedding is fed into an intra-graph GN block, followed by a series of graph pooling operators to construct the mega graph $\mathcal{MG}$ and features $(\mathbf{X}^0)^{\mathcal{MG}}$. In the *process* stage, an elementary component is a layer shaped like a mirrored $\mathbb{E}$, which is referred to as the `Mee`

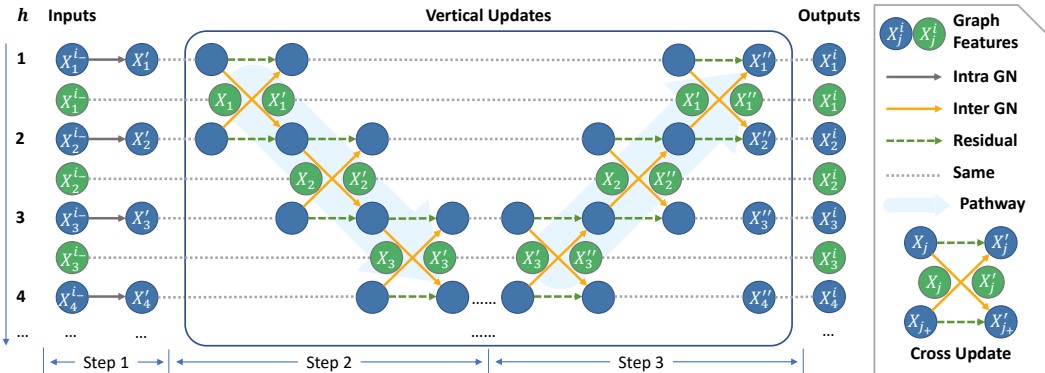

Figure 4: Illustration of the `Mee` layer, where $i_-$ means $i-1$ and $j_+$ means $j+1$. The blue and green circles represent features of intra- and inter-graphs, respectively. The grey and golden arrows indicate the intra and inter GN blocks, respectively. The cross update exchanges information between consecutive heights using inter GN blocks, detailed in the main text. There are three steps of updates, where the first step updates intra-graph features. The second step sequentially applies cross updates from lower to higher levels. The information is accumulated along the pathway and passes to higher levels. The procedure is reversed in the third step.

layer. The *process* module is composed by stacking the `Mee` layers for $n$ times. The $i$-th `Mee` layer takes the features $(\mathbf{X}^{i-1})^{\mathcal{MG}}$ as inputs and outputs $(\mathbf{X}^i)^{\mathcal{MG}}$ (with residual links (He et al., 2016)) by applying GN blocks in a designated order. In the *decode* stage, the features $(\mathbf{X}^n)^{\mathcal{MG}}$ are aggregated to task-dependent representations using readout functions.

**Mee Layer.** As we can observe in Fig. 3, the `Mee` layer contains horizontal flows at each height and vertical bidirectional pathways across multiple scaled graphs. A zoom-in structure of the `Mee` layer is depicted in Fig. 4. At each `Mee` layer, the messages are passed through one step horizontal flow, i.e., the features of intra-graphs are fed into a GN block at each height as shown in Step 1. Then, the messages propagate from height 1 to height $h$ and reverse back, where the features are updated through the intra-edges according to the arrows (called the cross update or XUPD) in Step 2 (height 1 to height $h$) and Step 3 (reverse back). Within a single `Mee` layer, the information can be efficiently exchanged across multiple scaled graphs. By stacking the `Mee` layer into a deeper architecture, the information exchange is repeated for $n$ times where $n$ is the number of stacked `Mee` layers.

Formally, let $(\mathbf{X}^{i-1})^{\mathcal{MG}} = \{(\mathbf{X}^{i-1})^{\mathcal{G}_{1:h}}, (\mathbf{X}^{i-1})^{\hat{\mathcal{G}}_{1:h}}\}$ be the inputs of the $i$-th `Mee` layer. For simplicity, we omit the superscript of graph identities and rewrite the features of intra- and inter-graphs as $\{\mathbf{X}_j^{i-1}\}_{j=1,\cdots,h} := (\mathbf{X}^{i-1})^{\mathcal{G}_{1:h}}$ and $\{\hat{\mathbf{X}}_j^{i-1}\}_{j=1,\cdots,h-1} := (\mathbf{X}^{i-1})^{\hat{\mathcal{G}}_{1:h}}$. Then, the updates in Step 1 can be written as $\mathbf{X}_j' = \text{GN}_{\text{intra}}^{i,j}(\mathcal{G}_j, \mathbf{X}_j^{i-1})$, where $\mathbf{X}_j'$ is the updated features of intra-graph $\mathcal{G}_j$. For the vertical updates in Steps 2 and 3, we define the cross update between consecutive heights $j$ and $j+1$ to be a function $(\mathbf{X}_j', \hat{\mathbf{X}}_j', \mathbf{X}_{j+1}') = \text{XUPD}(j, \mathbf{X}_j, \hat{\mathbf{X}}_j, \mathbf{X}_{j+1})$. This function is realized by first merging $\mathbf{X}_j$ (node-wisely) with $\hat{\mathbf{X}}_j$ as $\bar{\mathbf{X}}_j$, applying GN blocks on inter-graph $\hat{\mathcal{G}}_j$ by $\bar{\mathbf{X}}_j' = \text{GN}_{\text{inter}}^{i,j}(\hat{\mathcal{G}}_j, \bar{\mathbf{X}}_j)$, and finally retrieving $\mathbf{X}_j', \mathbf{X}_{j+1}'$ and $\hat{\mathbf{X}}_j'$ from $\bar{\mathbf{X}}_j'$. We denote this default realization as X-Conv. The cross update function can also be realized using the REDUCE and EXPAND operation of POOL (see Section 2.2) by $\mathbf{X}_{j+1}' = \text{REDUCE}(\hat{\mathcal{G}}_j, \hat{\mathbf{X}}_j^0, \mathbf{X}_j)$ and $\mathbf{X}_j' = \text{EXPAND}(\hat{\mathcal{G}}_j, \hat{\mathbf{X}}_j^0, \mathbf{X}_{j+1})$, where $\hat{\mathcal{G}}_j$ is the $j$-th inter-graph. We denote such realization as X-Pool, which is standard for most pooling methods. The intra and inter GN blocks can share parameters among all $j$'s that generalize to different heights, or among all $i$'s that generalize to different depths.

The outputs of the *process* stage are the updated features $\{\mathbf{X}_j^i\}_{j=1,\cdots,h}$ and $\{\hat{\mathbf{X}}_j^i\}_{j=1,\cdots,h-1}$. In our implementation, we add residual links from $\mathbf{X}_j^{i-1}$ to $\mathbf{X}_j^i$ and from $\hat{\mathbf{X}}_j^{i-1}$ to $\hat{\mathbf{X}}_j^i$ to provide shortcuts bypassing the entire `Mee` layer.

### 3.3 MODULE CHOICE AND INNOVATION

In `MeGraph`, there are two elementary modules, i.e., the graph pooling operator and the GN block. The `MeGraph` architecture can adopt any graph pooling method as long as it belongs to the POOL function family introduced in Section 2.2). Also, the choice of the GN block is not limited to the graph convolution layer as used in standard GCN, GIN or GAT.

**Graph Pooling.** There are a number of commonly used graph pooling methods, including Diff-Pool (Ying et al., 2018), TopKPool (Gao & Ji, 2019), EdgePool (Diehl et al., 2019), etc. Among

those, EdgePool is a promising method because it is trainable, sparse, and adaptable, according to the taxonomy proposed in Grattarola et al. (2022). It also preserves the connectivity in graphs, i.e., if two subsets of nodes $S_1$ and $S_2$ are connected in the input graph, the reduced nodes $\tilde{v}_1$ and $\tilde{v}_2$ are still connected in the pooled graph. However, the pooling ratio (the number of nodes after pooling over the number of nodes before pooling) for applying one EdgePool operator is lower bounded by $50\%$ and not adjustable since EdgePool seeks to only contract edges without overlapping connected nodes. This is inflexible when the original graph (of $N$ nodes) is extremely large, with at least $\log_2 N$ pooling operations to reduce to a single node. In this paper, we extend the EdgePool method to deal with arbitrary pooling ratios. We propose the Stridden EdgePool (S-EdgePool) with a variable pooling stride within the framework of the POOL function family introduced in Section 2.2. Moreover, we propose an efficient implementation of S-EdgePool (containing EdgePool as a special case) using the disjoint-set data structure (Galler & Fisher, 1964) below.

In the SELECT step, S-EdgePool shares the same computations as in EdgePool to generate learnable edge scores, as detailed in Appendix C.1.1. Then, we propose a clustering procedure to determine the subset of nodes to be reduced. Let $I_{\mathbf{v}}$ be the identifier of the cluster containing a set of nodes $\mathbf{v}$. Initially, we let $\mathbf{v} = \{v\}$ for every single node $v$. A contraction of an edge merges a pair of nodes $(v, v')$ connected by this edge (where $v \in \mathbf{v}$, $v' \in \mathbf{v}'$ and $\mathbf{v} \neq \mathbf{v}'$), and thus unifies the cluster identifiers, i.e., $I_{\mathbf{v}} = I_{\mathbf{v}'} = I_{\mathbf{v}_{\text{merge}}}$ and $\mathbf{v}_{\text{merge}} = \mathbf{v} \cup \mathbf{v}'$. That is, once an edge connecting any pair of nodes from two distinct clusters is contracted, we merge the two clusters and unify their identifiers. Edges are visited sequentially by a decreasing order on the edge scores, and contractions are implemented if valid. We set the maximum size of the node clusters to be a parameter $\tau_c$, where $\tau_c = 2$ degenerates to the case of EdgePool (Diehl et al., 2019). We further introduce the pooling ratio $\eta_v$ to control the minimal number of remaining clusters after edge contractions to be $N^v * \eta_v$. Contractions that violate the above two constraints are invalid and will be skipped. Both parameters control the number of nodes in the pooled graph. In our implementation, the cluster of nodes is dynamically maintained using the disjoint-set data structure (Galler & Fisher, 1964). The detailed procedures of CONNECT, REDUCE and EXPAND functions are provided in Appendix C.1.2. The pseudocode of the entire algorithm is given in Algorithm 2 of Appendix C.1.3.

**GN block.** In this paper, we realize the full GN block (introduced in Section 2.1) as a graph full network (GFuN) layer. A practical difference from the full GN block in (Battaglia et al., 2018) is that we deactivate some links in the full GN block to reduce the computational complexity and the number of parameters. More details are available in Appendix C.2. We use GFuN as the basic component because of its excellent flexibility. We compare it with GCN in Appendix E.2.

**Encoder and decoder.** The MeGraph model can choose most input embedding methods (including positional encodings) and readout functions used in GNNs. Details are in Appendix C.3.

### 3.4 COMPUTATIONAL COMPLEXITY

The overall complexity of the MeGraph model depends on the height $h$, the number of Mee layers $n$, and the choices of the modules, as well as the corresponding hyper-parameters.

Let $D$ be the embedding size, $V$ be the number of nodes, and $E$ be the number of edges in the input graph $\mathcal{G}$. The time complexity of S-Edgepool is $O(ED+E\log E)$, where $O(ED)$ is the complexity of computing edge scores and $O(E\log E)$ comes from sorting the edge scores. The dynamic node clustering using disjoint-set is of $O(E\alpha(E))$ complexity where $\alpha(E)$ is a function that grows slower than $\log(E)$ (Tarjan & Van Leeuwen, 1984). The time complexity of a GFuN layer is $O(VD^2+ED)$. For simplicity, we assume both the pooling ratios of nodes and edges are $\eta$. Then, the total time complexity to build the mega graph $\mathcal{MG}$ is $O((ED + E\log E)/(1 - \eta))$, where $\sum_{i=0}^{h-1} \eta^i < 1/(1 - \eta)$. Similarly, the total time complexity of an Mee layer is $O((VD^2 + ED)/(1 - \eta))$, which is the same as a normal GNN layer if we regard $1/(1 - \eta)$ as a constant (e.g., it is a constant of 2 when $\eta = 0.5$). In practice, we introduce some variants of MeGraph to further reduce the time complexity in Appendix C.4.

### 4 EXPERIMENTS

We conduct comprehensive experiments on both node and graph prediction tasks across a large variety of synthetic and real-world datasets to show the superior performance of the MeGraph model. We also demonstrate the importance of introducing S-EdgePool by ablation studies. Due to space limitations, statistics of the datasets are provided in Appendix B.1, and the training and implementation details are reported in Appendix D.

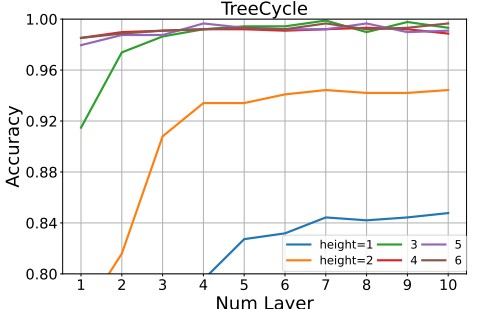 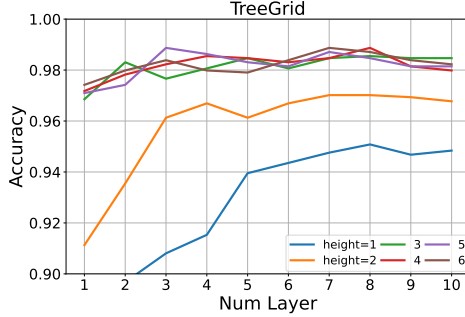

Figure 5: Node classification accuracy (averaged over 10 random repetitions) for `MeGraph` on TreeCycle (left) and TreeGrid (right) datasets by varying the height $h$ and the number of `Mee` layers $n$. Clear gaps can be observed among heights 1, 2, and $\geq 3$. Detailed numbers can be found in Table 10 of Appendix E.3.

## 4.1 BASELINES

The height $h$ is the key parameter determining the overall architecture of the `MeGraph` model. When we set $h = 1$, the `MeGraph` model reduces to a normal GNN over the original graph, which will be treated as a reliable baseline. To show the importance of repeated multi-scale information exchange, we also compare with the baseline method by setting the number of `Mee` layers $n = 1$. We also compare with the Graph U-Nets (Gao & Ji, 2019), which is approximately equivalent to a U-Shaped variant under the `MeGraph` architecture. We use the GFuN layer as the core GN block in these models.

## 4.2 SYNTHETIC DATASETS

We first study the effect of the max height $h$ and the number of `Mee` layers $n$ on 4 synthetic datasets introduced by Ying et al. (2019), including the BAShape, BACommunity, TreeCycle and TreeGrid. Each dataset contains one graph formed by attaching multiple motifs to a base graph. The motif can be a 'house'-shaped network (BAShape, BACommunity), six-node cycle (TreeCycle), or 3-by-3 grid (TreeGrid). The task is to identify the nodes of the motifs in the fused graph.

The results of TreeCycle and TreeGrid are shown in Fig. 5. We can observe clear gaps among curves of $h = 1$, $h = 2$ and $h \geq 3$ for all values of $n$. This indicates that $h$ is crucially important for recognizing the motifs. Similar conclusions can also be drawn in the easier datasets BAShape and BACommunity (Fig. 6 in Appendix E.3).

## 4.3 GRAPH THEORY BENCHMARK

An important benefit of the MeGraph architecture is that it facilitates long-distance inference over graphs. To verify this, we create a graph theory benchmark containing 3 graph regression tasks and 2 node regression tasks, for solving which long-distance inference is necessary. The graph regression tasks include Single Source Single Destination Shortest Path ($\text{SP}_{\text{sssd}}$), Maximum Connected Component of the same color (MCC) and Graph Diameter (Diameter). The node regression tasks are Single Source Shortest Path ($\text{SP}_{\text{ss}}$) and Eccentricity of nodes (ECC). All these problems are based on artificially generated graphs. Following Corso et al. (2020), we use their methods to generate undirected and unweighted graphs randomly. In addition, we propose three new methods: cycle, pesudotree and geographic threshold graphs. We create a dataset for each task and each graph generation method, resulting in a total of 55 datasets after filtering out the trivial cases. The details of those tasks and dataset generation can be found in Appendix B.2. As shown in Table 1, the `MeGraph` model achieves significantly smaller regression loss compared with all the baselines ($h = 1$, $n = 1$ and U-Shaped net), even when the baseline methods take more GNN layers.

**Ablation Studies.** We first vary the node pooling ratio $\eta_v$ and the maximum cluster size $\tau_c$ of S-EdgePool to evaluate the performance. The advantage of a flexible pooling stride is shown in the lower part of Table 1, where the best one achieves almost 4x smaller error ([$\eta_v = 0.3, \tau_c = 4$] 0.624 vs. [$\tau_c = 2$] 2.337). We further experiment with the X-Pool (see the definition of XUPD in Sec 3.2) variation. The dropped performance ([X-Pool] 1.165 vs. [X-Conv] 0.624) indicates the importance of using GN block for inter-graph updates.

## 4.4 REAL WORLD DATASETS

**Experimental Protocol.** In this subsection, we evaluate `MeGraph` on public real-world graph benchmarks. To fairly compare `MeGraph` with the baselines, we use the following experimental protocols. We first report the public baseline results and our reproduced standard GCN's results. We

Table 1: Results on Graph Theory Benchmark. For each task, we report the MSE regression loss on test set, averaged over different graph generation methods. Darker blue cells denote better performance. We refer more details to Appendix E.4.

| Category | Model | $SP_{sssd}$ | MCC | Diameter | $SP_{ss}$ | ECC | Average |
|---|---|---|---|---|---|---|---|
| Baselines ($h$=1) | $n$=1 | 12.188 | 1.377 | 12.654 | 25.159 | 21.522 | 13.680 |
| | $n$=5 | 4.246 | 1.093 | 6.048 | 13.715 | 20.287 | 8.821 |
| | $n$=10 | 2.488 | 1.119 | 5.812 | 7.819 | 20.201 | 7.481 |
| MeGraph EdgePool ($h$=5, $\tau_c$=2) | $n$=9 (U-Shaped) | 2.683 | 1.144 | 5.680 | 2.801 | 20.102 | 6.656 |
| | $n$=1 | 1.856 | 0.772 | 4.801 | 6.110 | 14.920 | 5.662 |
| | $n$=5 | 0.817 | 0.616 | 2.196 | 0.785 | 6.892 | 2.337 |
| MeGraph S-EdgePool Ablation ($h$=5, $n$=5) | $\tau_c$=3 | 0.648 | 0.600 | 0.575 | 0.501 | **0.856** | 0.644 |
| | $\eta_v$=0.3 | 2.331 | 0.583 | 0.964 | 3.984 | 2.021 | 1.840 |
| | $\eta_v$=0.3, $\tau_c$=4 | **0.584** | **0.565** | **0.517** | **0.475** | 0.925 | **0.624** |
| | $\eta_v$=0.5, $\tau_c$=4 | 1.103 | 0.600 | 0.835 | 1.331 | 2.016 | 1.163 |
| | $\eta_v$=0.3, $\tau_c$=4 (X-Pool) | 0.935 | 0.619 | 0.734 | 1.618 | 2.014 | 1.165 |

Table 2: Results on GNN benchmark. † denotes the results are reported in (Dwivedi et al., 2020). Regression tasks are colored with *blue*. ↓ indicates that smaller numbers are better. Results of classification tasks are colored with *green*. ↑ indicates that larger numbers are better. Darker colors indicate better performance.

| Model | ZINC ↓ | AQSOL ↓ | MNIST ↑ | CIFAR10 ↑ | PATTERN ↑ | CLUSTER ↑ |
|---|---|---|---|---|---|---|
| GCN[†] | 0.416 ±0.006 | 1.372 ±0.020 | 90.120 ±0.145 | 54.142 ±0.394 | 85.498 ±0.045 | 47.828 ±1.510 |
| GIN[†] | 0.387 ±0.015 | 1.894 ±0.024 | 96.485 ±0.252 | 55.255 ±1.527 | 85.590 ±0.011 | 58.384 ±0.236 |
| GAT[†] | 0.475 ±0.007 | 1.441 ±0.023 | 95.535 ±0.205 | 64.223 ±0.455 | 75.824 ±1.823 | 57.732 ±0.323 |
| GatedGCN[†] | 0.435 ±0.011 | 1.352 ±0.034 | 97.340 ±0.143 | 67.312 ±0.311 | 84.480 ±0.122 | 60.404 ±0.419 |
| GCN | 0.426 ±0.015 | 1.397 ±0.029 | 90.140 ±0.140 | 51.050 ±0.390 | 84.672 ±0.054 | 47.541 ±0.940 |
| MeGraph ($h$=1) | 0.323 ±0.002 | 1.075 ±0.007 | 97.570 ±0.168 | 69.890 ±0.209 | 84.845 ±0.021 | 58.178 ±0.079 |
| MeGraph ($n$=1) | 0.310 ±0.005 | 1.038 ±0.018 | 96.867 ±0.167 | 68.522 ±0.239 | 85.507 ±0.402 | 50.396 ±0.082 |
| MeGraph | 0.260 ±0.005 | **1.002** ±0.021 | **97.860** ±0.098 | **69.925** ±0.631 | 86.507 ±0.067 | 68.603 ±0.101 |
| MeGraph$_{best}$ | **0.202** ±0.007 | **1.002** ±0.021 | **97.860** ±0.098 | **69.925** ±0.631 | **86.732** ±0.023 | **68.610** ±0.164 |

then replace GCN layers with GFuN layers (which is equivalent to MeGraph ($h = 1$)) to serve as another baseline. We tune the hyper-parameters (such as learning rate, dropout rate and the readout global pooling method, etc.) of MeGraph ($h = 1$) and choose the best configurations. We then run other diversely configured MeGraph candidates by tuning other hyper-parameters that only matters for $h > 1$, and these hyper-parameters are referred to as the MeGraph hyper-parameters. MeGraph ($n = 1$) also serves as a baseline method, which does not enables repeated information exchange. The standard MeGraph uses an uniform hyper-parameter setting for all the datasets. We also report the best performance of MeGraph with specifically tuned hyper-parameters in each dataset, denoted as MeGraph$_{best}$. Detailed configurations are put in Table 5 in the Appendix.

**GNN Benchmark** (Dwivedi et al., 2020). We experiment on three types of GNN benchmark datasets, which are chemical data (ZINC and AQSOL), image data (MNIST and CIFAR10) and social network data (PATTERN and CLUSTER). The tasks are regressing certain properties of molecule graphs (graph regression), classifying the super-pixel graphs (graph classification), and recognizing the patterns of nodes or clustering nodes (node classification), respectively. More details can be found in their original works. As shown in Table 2, Megraph outperforms the public results reported in (Dwivedi et al., 2020) and our two baselines.

**Open Graph Benchmark (OGB)** (Hu et al., 2020). We choose 10 datasets related to molecular graphs from the graph prediction tasks of OGB, 7 out of which are classification tasks (molhiv, molbace, molbbbp, molclintox, molsider, moltox21 and moltoxcast) and the others are regression tasks (molesol, molfreesolv and mollipo). For all datasets, each graph represents a molecular compound. The node features are properties of atoms and the edge features are properties of bonds between atoms. The task of all datasets is to predict some properties of molecule graphs based on their chemical structures. As shown in Table 3, Megraph achieves $1\%$ to $3\%$ absolute gains on classification tasks, and about $10\%$ relative gains on regression tasks compared to the baseline MeGraph ($h = 1$).

**TU Datasets** (Morris et al., 2020). The results of 10 popular TU datasets are put in Appendix E.1.

## 5 RELATED WORKS

**Feature Pyramids and Multi-Scale Feature Fusion.** Multi-scale feature fusion methods on image feature pyramids have been widely studied in computer vision literature, including the U-Net (Ronneberger et al., 2015), FPN (Lin et al., 2017), UNet++ (Zhou et al., 2018), and some recent ap-

Table 3: Results on OGB-G. † indicates that the results are reported in (Hu et al., 2020).

| Model | molhiv ↑ | molbace ↑ | molbbbp ↑ | molclintox ↑ | molsider ↑ |
|---|---|---|---|---|---|
| GCN[†] | 76.06 ±0.97 | 79.15 ±1.44 | 68.87 ±1.51 | 91.30 ±1.73 | 59.60 ±1.77 |
| GIN[†] | 75.58 ±1.40 | 72.97 ±4.00 | 68.17 ±1.48 | 88.14 ±2.51 | 57.60 ±1.40 |
| GCN | 75.40 ±1.29 | 76.01 ±3.31 | 67.35 ±0.96 | 89.62 ±2.27 | 58.08 ±0.78 |
| MeGraph ($h$=1) | 78.54 ±1.14 | 71.77 ±2.15 | 67.56 ±1.11 | 89.77 ±3.48 | 58.28 ±0.51 |
| MeGraph ($n$=1) | 78.56 ±1.02 | 79.72 ±1.24 | 67.34 ±0.98 | 91.07 ±2.21 | 58.08 ±0.59 |
| MeGraph | 77.20 ±0.88 | 78.52 ±2.51 | **69.57** ±2.33 | 92.04 ±2.19 | 59.01 ±1.45 |
| MeGraph$_{best}$ | **79.20** ±1.80 | **83.52** ±0.47 | **69.57** ±2.33 | **92.06** ±1.32 | **63.43** ±1.10 |

| Model | moltox21 ↑ | moltoxcast ↑ | molesol ↓ | molfreesolv ↓ | mollipo ↓ |
|---|---|---|---|---|---|
| GCN[†] | 75.29 ±0.69 | 63.54 ±0.42 | 1.114 ±0.03 | 2.640 ±0.23 | 0.797 ±0.02 |
| GIN[†] | 74.91 ±0.51 | 63.41 ±0.74 | 1.173 ±0.05 | 2.755 ±0.34 | 0.757 ±0.01 |
| GCN | 75.11 ±0.41 | 64.13 ±0.52 | 1.141 ±0.02 | 2.407 ±0.15 | 0.788 ±0.01 |
| MeGraph ($h$=1) | 75.89 ±0.45 | 64.49 ±0.46 | 1.079 ±0.02 | 2.017 ±0.08 | 0.768 ±0.00 |
| MeGraph ($n$=1) | 77.01 ±0.93 | 66.89 ±1.21 | 0.896 ±0.04 | 1.892 ±0.06 | 0.730 ±0.01 |
| MeGraph | **78.11** ±0.47 | 67.67 ±0.53 | 0.886 ±0.02 | **1.876** ±0.05 | 0.726 ±0.00 |
| MeGraph$_{best}$ | **78.11** ±0.47 | **67.90** ±0.19 | **0.867** ±0.02 | **1.876** ±0.05 | **0.688** ±0.01 |

proaches (Yu et al., 2018; Liu et al., 2018; Lin et al., 2019; Li et al., 2020). HRNet (Wang et al., 2020) is a similar method compared to MeGraph. HRNet alternates between multi-resolution convolutions and multi-resolution fusion by stridden convolutions. However, the above methods are developed for image data, and key differences compared to these approaches is that the multi-scale feature fusion in MeGraph is incorporated with the inter-graphs generated by graph pooling, and the fusion process is repeated for multiple times. For graph networks, the GraphFPN (Zhao et al., 2021) builds mappings between the image and graph feature pyramids according to the superpixel hierarchy, and it applies GNN layers on the hierarchical graph to exchange information within the graph pyramid. However, the flow of inference still propagates for a single pass over a fixed contextual-hierarchical-contextual structure (see Fig. 1 of (Zhao et al., 2021)). Gao & Ji (2019) and Fey et al. (2020) have also explored similar ideas in graph structured data. Our approach shares the general idea of multi-scale information fusion, but it is the first method that builds a mega architecture with graph pooling and GN blocks that achieve efficient multi-scale information exchange in the domain of graph representation learning.

**Graph Pooling Methods.** Graph pooling is an important part in hierarchical graph representation learning. There have been some traditional graph pooling methods like METIS (Karypis & Kumar, 1998) in early literature. Recently, many learning based graph pooling methods have been proposed, including the DiffPool (Ying et al., 2018), TopKPool (Gao & Ji, 2019), SAG pool (Lee et al., 2019), EdgePool (Diehl et al., 2019), MinCutPool (Bianchi et al., 2020), and MEWISPool (Nouranizadeh et al., 2021), etc. In MeGraph, we generalize the EdgePool method as S-EdgePool to build the mega graph, while this operator can be switched to any one of the above mentioned pooling method.

**Graph Neural Network (GNN) Layers.** The GNN layer is the core module of graph representation learning models. Typical GNNs include the GCN (Kipf & Welling, 2016), GraphSage (Hamilton et al., 2017), GAT (Veličković et al., 2018; Brody et al., 2021), GIN (Xu et al., 2018), PNA (Corso et al., 2020). MeGraph adopts the full GN block (Battaglia et al., 2018) by removing part of links in the module as an elementary block, and similarly this can be replaced by any one of the popular GNN blocks.

## 6 LIMITATIONS AND FUTURE WORK

The MeGraph model suffers from some limitations. The introduced mega graph architecture inevitably increases both the number of trainable parameters and tuneable hyper-parameters. The flexible choices of many modules in MeGraph post burdens on tuning the architecture on specific datasets. For future research, MeGraph encourages new graph pooling methods to yield edge features in addition to node features, when mapping the input graph to the pooled graph. It is also possible to improve MeGraph using adaptive computational steps (Tang et al., 2020). Another direction is to apply some expressive models like Transformers (Vaswani et al., 2017) and Neural Logic Machines (Dong et al., 2018; Xiao et al., 2022) (only) over the pooled small-sized graphs, since these models are computational expensive.

## 7 REPRODUCIBILITY STATEMENT

We will post an anonymous code repository link in an official comment on Openreview. We set the random seed as 2022 for all experiments to enable reproducible results. We provide dataset statistics in Table 4 and details for the proposed graph theory benchmark in Appendix B.2. All details of the hyper-parameters are reported in Table 5. Configuration of all hyper-parameters and the command lines to reproduce the experiments will be included in the code repository.

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

## A  SUMMARY OF APPENDIX

We present dataset details in Section B, method details in Section C, implementation and training details in Section D and extra experiment results in Section E.

## B  DATASET DETAILS

### B.1  DATASET STATISTICS AND METRICS

We provide the statistics of all datasets used in our experiments in Table 4 and introduce the evaluation metrics for each dataset.

For Synthetic datasets, we use classification accuracy (ACC) as the evaluation metric. We use Mean Square Error (MSE) as the evaluation metric for all datasets in our Graph Theory Benchmark. For GNN Benchmark, we follow the original work (Dwivedi et al., 2020) for evaluation, i.e., Mean Absolute Error (MAE) for ZINC and AQSOL, classification accuracy for MNIST and CIFAR10, and balanced classification accuracy for PATTERN and CLUSTER. For OGB Benchmark, we follow the original work (Hu et al., 2020) and use the ROC-AUC for classification tasks and Root Mean Square Error (RMSE) for regression tasks. For TU datasets, we follow the setting used by (Chen et al., 2019) and use classification accuracy as the evaluation metric.

### B.2  GRAPH THEORY BENCHMARK

In this section, we provide the details about the tasks and how the graph features and the labels are generated given a base graph $\mathcal{G} = (\mathcal{V}, \mathcal{E})$:

- Single source single destination shortest path ($SP_{sssd}$): a source node $s \in \mathcal{V}$ and a destination node $t \in \mathcal{V}$ are selected uniform randomly. The feature of each node $v$ contains three numbers: (1, whether the node $v$ is $s$, whether the node $v$ is $t$). The label of a graph is the length of the shortest path from $s$ to $t$.

- A maximum connected component of the same color (MCC): each node of the graph is colored with one of three colors. The feature for each node is the one-hot representation of its color. The label of graph is the size of the largest connected component of the same color for each color.

- Graph diameter (Diameter): the label of the graph is the diameter of the graph. The diameter of a graph $\mathcal{G}$ is the maximum of the set of shortest path distances between all pairs of nodes in the graph. The feature of each node is a uniform number 1.

- Single source shortest path ($SP_{ss}$): a source node $s$ is selected uniformly randomly. The feature of each node contains two numbers: (1, whether the node is $s$). The label of each node is the length of the shortest path from $s$ to this node.

- Graph eccentricity (ECC): the label of each node $v$ is node's eccentricity in the graph, which is the maximum distance from $v$ to the other nodes. The feature of each node is a uniform number 1.

For each task and graph generation method, We generate the dataset by the following steps:

- Sample $N$ (number of nodes) from $[20, 50]$, totally 300 graphs. These numbers can be configured.
- Use the graph generation method to generate a graph of $N$ nodes.
- Create graph features and labels according to the task.

We then provide the details about the random graph generation methods we used to create our Graph Theory datasets.

Following Corso et al. (2020), we continue to use undirected and unweighted graphs from a wide variety of types. We inherit their 10 random graph generation methods and quote their descriptions here for completeness (the percentage after the name is the approximate proportion of such graphs in the mixture setting).

Table 4: The statistics of the datasets used in experiments. Some statistics (like the average number of edges) of the Graph Theory datasets may vary depending on different random graph generation methods. The regression tasks are marked with ✓ in a separate column. The tasks of 4 synthetic datasets are transductive, where the same graph is used for both training and testing. We do not use the node labels as features during the training time. The train-val-test split is over nodes. All other datasets in the table are inductive, where the testing graphs do not occur during training, and the train-val-test split is over graphs.

| Collection | Dataset | # Graphs | Avg # Nodes | Avg # Edges | # Node Feat | # Edge Feat | # Classes | Task | Reg. |
|---|---|---|---|---|---|---|---|---|---|
| Synthetic | BaShape | 1 | 700 | 1761 | 1 | - | 4 | Trans-Node | |
| Synthetic | BaCommunity | 1 | 1400 | 3872 | 10 | - | 8 | Trans-Node | |
| Synthetic | TreeCycle | 1 | 871 | 970 | 1 | - | 2 | Trans-Node | |
| Synthetic | TreeGrid | 1 | 1231 | 1705 | 1 | - | 2 | Trans-Node | |
| GraphTheory | $SP_{sssd}$ | 300 | 35.0 | - | 3 | - | - | Graph | ✓ |
| GraphTheory | Diameter | 300 | 35.0 | - | 1 | - | - | Graph | ✓ |
| GraphTheory | MCC | 300 | 35.0 | - | 3 | - | - | Graph | ✓ |
| GraphTheory | $SP_{ss}$ | 300 | 35.0 | - | 2 | - | - | Node | ✓ |
| GraphTheory | ECC | 300 | 35.0 | - | 1 | - | - | Node | ✓ |
| GNNBenchmark | ZINC | 12000 | 23.16 | 49.83 | 28 | 4 | 2 | Graph | ✓ |
| GNNBenchmark | AQSOL | 9823 | 17.57 | 35.76 | 65 | 5 | 2 | Graph | ✓ |
| GNNBenchmark | MNIST | 70000 | 70.57 | 564.53 | 3 | 1 | 10 | Graph | |
| GNNBenchmark | CIFAR10 | 60000 | 117.63 | 941.07 | 5 | 1 | 10 | Graph | |
| GNNBenchmark | PATTERN | 14000 | 118.89 | 6078.57 | 3 | - | 2 | Node | |
| GNNBenchmark | CLUSTER | 12000 | 117.20 | 4301.72 | 7 | - | 6 | Node | |
| OGB Graph | molhiv | 41127 | 25.51 | 80.45 | 9 | 3 | 2 | Graph | |
| OGB Graph | molbace | 1513 | 34.09 | 107.81 | 9 | 3 | 2 | Graph | |
| OGB Graph | molbbbp | 2039 | 24.06 | 75.97 | 9 | 3 | 2 | Graph | |
| OGB Graph | molclintox | 1477 | 26.16 | 81.93 | 9 | 3 | 2 | Graph | |
| OGB Graph | molsider | 1427 | 33.64 | 104.36 | 9 | 3 | 2 | Graph | |
| OGB Graph | moltox21 | 7831 | 18.57 | 57.16 | 9 | 3 | 2 | Graph | |
| OGB Graph | moltoxcast | 8576 | 18.78 | 57.30 | 9 | 3 | 2 | Graph | |
| OGB Graph | molesol | 1128 | 13.29 | 40.64 | 9 | 3 | - | Graph | ✓ |
| OGB Graph | molfreesolv | 642 | 8.7 | 25.50 | 9 | 3 | - | Graph | ✓ |
| OGB Graph | mollipo | 4200 | 27.04 | 86.04 | 9 | 3 | - | Graph | ✓ |
| TU | MUTAG | 188 | 17.93 | 19.79 | 7 | - | 3 | Graph | |
| TU | NCI1 | 4110 | 29.87 | 32.30 | 37 | - | 2 | Graph | |
| TU | PROTEINS | 1113 | 39.06 | 72.82 | 4 | - | 2 | Graph | |
| TU | D&D | 1178 | 284.32 | 715.66 | 89 | - | 2 | Graph | |
| TU | ENZYMES | 600 | 32.63 | 62.14 | 21 | - | 6 | Graph | |
| TU | IMDB-B | 1000 | 19.77 | 96.53 | 10 | - | 2 | Graph | |
| TU | IMDB-M | 1500 | 13.00 | 65.94 | 10 | - | 3 | Graph | |
| TU | RE-B | 2000 | 429.63 | 497.75 | 10 | - | 2 | Graph | |
| TU | RE-M5K | 4999 | 508.52 | 594.87 | 10 | - | 5 | Graph | |
| TU | RE-M12K | 11929 | 391.41 | 456.89 | 10 | - | 11 | Graph | |

- **Erdös-Rényi (ER)** (20%) (Erdős et al., 1960): with a probability of presence for each edge equal to $p$, where $p$ is independently generated for each graph from $\mathcal{U}[0, 1]$

- **Barabási-Albert (BA)** (20%) (Albert & Barabási, 2002): the number of edges for a new node is $k$, which is taken randomly from $\{1, 2, \ldots, N - 1\}$ for each graph

- **Grid** (5%): $m \times k$ 2d grid graph with $N = mk$ and $m$ and $k$ as close as possible

- **Caveman** (5%) (WATTS, 2003): with $m$ cliques of size $k$, with $m$ and $k$ as close as possible

- **Tree** (15%): generated with a power-law degree distribution with exponent 3

- **Ladder graphs** (5%)

- **Line graphs** (5%)

- **Star graphs** (5%)

- **Caterpillar graphs** (10%): with a backbone of size $b$ (drawn from $\mathcal{U}[1, N)$), and $N - b$ pendent vertices uniformly connected to the backbone

- **Lobster graphs** (10%): with a backbone of size $b$ (drawn from $\mathcal{U}[1, N)$), $p$ (drawn from $\mathcal{U}[1, N - b]$) pendent vertices uniformly connected to the backbone, and additional $N - b - p$ pendent vertices uniformly connected to the previous pendent vertices.

Additional, we add three more graph generation methods:

- **Cycle graphs**

- **Pseudotree graphs**: A tree graph plus an additional edge. The graph is generated by first generating a cycle graph of size $m = \mathrm{sample}(0.3N, 0.6N)$. Then $n - m$ remaining nodes are sampled to $m$ parts, where $i$-th part represents the size of the tree hanging on the $i$-th node on the cycle. The trees are randomly generated with the given size.

- **Geographic (Geo) graphs**: geographic threshold graphs, but with added edges via a minimum spanning tree algorithm, to ensure all nodes are connected. This graph generation method is introduced by Battaglia et al. (2018) in their codebase [1]. We use the geographic threshold $\theta = 200$ instead of the default value $\theta = 1000$.

Note that we do not have randomization after the graph generation as in Corso et al. (2020). Therefore, very long diameter is preserved for some type of graphs.

## C  METHOD DETAILS

### C.1  S-EDGEPOOL

#### C.1.1  EDGE SCORE GENERATION

Both S-EdgePool and EdgePool methods compute a raw edge score $\mathbf{r}_k$ for each edge $k$ using a linear layer:

$$\mathbf{r}_k = \mathbf{W} \cdot (\mathbf{V}_{s_k} || \mathbf{V}_{t_k} || \mathbf{E}_k) + \mathbf{b}$$

where $s_k$ and $t_k$ are the source and target nodes of edge $k$, $\mathbf{V}$ is node features, $\mathbf{E}$ is edge features, $\mathbf{W}$ and $\mathbf{b}$ are learned parameters. The raw edge scores are further normalized by a local softmax function over all edges of a node:

$$\mathbf{w}_k = \exp(\mathbf{r}_k) / \sum_{k', t_{k'} = t_k} \exp(\mathbf{r}_{k'}),$$

and biased by a constant $0.5$ (Diehl et al., 2019).

---

[1]https://github.com/deepmind/graph_nets, the shortest path demo

### C.1.2 CONNECT, REDUCE AND EXPAND

In this subsection, we give the details of CONNECT, REDUCE and EXPAND functions of S-EdgePool.

The CONNECT function rebuilds the edge set $\tilde{\mathcal{E}}$ between the nodes in $\tilde{\mathcal{V}}$. As aforementioned, we build the pooled graph's nodes according to node clusters. We call this mapping function from node clusters to new nodes $n$. After that, we build the pooled graph's edges following three steps: First, for all edges in the original graph, we find out the corresponding node cluster(s) of its two endpoints (using a disjoint-set's find index operation). Then, we find out the corresponding new nodes by using the mapping function $n$. Last, we add a new edge between the new nodes.

In our experiments, the REDUCE and EXPAND we used are generalized from the method mentioned in Diehl et al. (2019). The REDUCE function computes new node features and edge features We follow their method to compute new node features by taking the sum of the node features and multiplying it by the edge score. Specifically, we generalize the computation between two nodes to a node cluster. The node clusters are maintained with a disjoint-set data structure. A cluster is consist of $|S_{\tilde{v}}|$ nodes. We define $\mathcal{E}_{\tilde{v}}^{ds}$ as a set of $|S_{\tilde{v}}| - 1$ edges, where the edges are the selected edges to be contracted in SELECT function and they connecting all nodes in a node cluster $S_{\tilde{v}}$.

$$c_{\tilde{v}} = \frac{1 + \sum_{e_k \in \mathcal{E}_{\tilde{v}}^{ds}} \mathbf{w}_k}{|S_{\tilde{v}}|}$$

$$\mathbf{V}_{\tilde{v}} = c_{\tilde{v}} \sum_{v \in S_{\tilde{v}}} \mathbf{V}_v$$

To integrate the edge features between two node clusters, we first find all the connected edges between the two node clusters (the edges between node clusters are edges that connect two nodes from different node clusters). Then, we use the sum of all the connected edges' features between the two node clusters as the new edge's features.

The term EXPAND we used refers an unpooling operation. When unpooling, we create an inverse mapping of pooled nodes to unpooled nodes.

$$\mathbf{V} = \frac{\tilde{V}_v}{\mathbf{w}_{avg}}$$

### C.1.3 PSEUDO CODE

The pseudo-code includes two parts, where Algorithm 1 describes how to maintain the clusters using a disjoint-set data structure, and Algorithm 2 describes the procedure of S-EdgePool that generates a pooled graph $\tilde{\mathcal{G}}$ with configurable node pooling ratio $\eta_v$ and maximum of cluster sizes $\tau_c$.

### C.2 GFUN

We first realize the $\phi_e$, $\phi_v$, $\phi_u$ functions in the full GN block (Sec 2.1 and (Battaglia et al., 2018)) as neural networks:

$$\mathbf{E}_k' = \mathrm{NN}_e(\mathbf{E}_k, \mathbf{V}_{s_k}, \mathbf{V}_{t_k}, \mathbf{u}), \tag{2}$$
$$\mathbf{V}_i' = \mathrm{NN}_v(\bar{\mathbf{E}}_i', \mathbf{V}_i, \mathbf{u}), \tag{3}$$
$$\mathbf{u}' = \mathrm{NN}_u(\bar{\mathbf{E}}', \bar{\mathbf{V}}', \mathbf{u}), \tag{4}$$

respectively, where

$$\bar{\mathbf{E}}_i' = \rho^{e \to v}(\{\mathbf{E}_k'\}_{k \in [1...N^e], t_k = i}), \tag{5}$$
$$\bar{\mathbf{E}}' = \rho^{e \to u}(\mathbf{E}'), \tag{6}$$
$$\bar{\mathbf{V}}' = \rho^{v \to u}(\mathbf{V}'). \tag{7}$$

We further decompose the neural networks according to the features in the function:

$$\mathrm{NN}_e(\mathbf{E}_k, \mathbf{V}_{s_k}, \mathbf{V}_{t_k}, \mathbf{u}) = \mathrm{NN}_{e \leftarrow e}(\mathbf{E}_k) + \mathrm{NN}_{e \leftarrow v_s}(\mathbf{V}_{s_k}) + \mathrm{NN}_{e \leftarrow v_t}(\mathbf{V}_{t_k}) + \mathrm{NN}_{e \leftarrow u}(\mathbf{u}), \tag{8}$$
$$\mathrm{NN}_v(\bar{\mathbf{E}}_i', \mathbf{V}_i, \mathbf{u}) = \mathrm{NN}_{v \leftarrow e}(\bar{\mathbf{E}}_i') + \mathrm{NN}_{v \leftarrow v}(\mathbf{V}_i) + \mathrm{NN}_{v \leftarrow u}(\mathbf{u}), \tag{9}$$
$$\mathrm{NN}_u(\bar{\mathbf{E}}', \bar{\mathbf{V}}', \mathbf{u}) = \mathrm{NN}_{u \leftarrow e}(\bar{\mathbf{E}}') + \mathrm{NN}_{u \leftarrow v}(\bar{\mathbf{V}}') + \mathrm{NN}_{u \leftarrow u}(\mathbf{u}) \tag{10}$$

---

**Algorithm 1** Get Cluster Index And Cluster Size of a Node (Using disjoint-set data structure)

---

**function** INITIALIZE DISJOINT SET(graph $\mathcal{G}(\mathcal{V}, \mathcal{E})$)
    **for** $v \in \mathcal{V}$ **do**
        $index[v] = v$                          ▷ the identifier of the cluster the node $v$ belongs to
        $size[v] = 1$
    **end for**
**end function**
**function** FIND INDEX(node $v$)
    **if** $index[v] = v$ **then**
        **return** $v$
    **else**
        $index[v] \leftarrow$ FIND INDEX($index[v]$)
        **return** $index[v]$
    **end if**
**end function**
**function** FIND INDEX AND SIZE(node $v$)
    $i \leftarrow$ FIND INDEX($v$)
    $s \leftarrow size[i]$
    **return** $i, s$
**end function**
**function** MERGE(cluster index $x$ and cluster index $y$)
    $size[y] \leftarrow size[x] + size[y]$
    $index[x] \leftarrow index[y]$
**end function**

---

However, such GN block uses 10 times number of parameters as the standard GCN (Kipf & Welling, 2016) layer when the node, edge and global embedding dimensions are all equivalent. In practice, we disable all computations related to global features $\mathbf{u}$, as well as the neural networks $\text{NN}_{e \leftarrow e}$ and $\text{NN}_{e \leftarrow v_t}$. We also set $\text{NN}_{v \leftarrow e}$ to be Identity.

In practice, we use summation function as the aggregator function $\rho^{e \rightarrow v}$ by default. But other choices like MEAN, MAX, gated summation, attention or their combinations can also be used.

Overall, we call such GN block as graph full network (GFuN) and use the practical setting in our experiments.

### C.3 ENCORDER AND DECODER

**Encoder**. For input embedding, we use Linear layer or Embedding layer to embed input features. For example, we follow Dwivedi et al. (2020) and use Linear layer on MNIST and CIFAR10 datasets, use Embedding layer on ZINC and AQSOL datasets. For molecular graph in OGB, we use the same embedding method as in the original work (Hu et al., 2020). Besides, we can adopt positional encoding methods like Laplacian (Dwivedi et al., 2020) and Random Walk (Dwivedi et al., 2021) to further embed global and local graph structure information. The embedding of positional encoding can be combined into (like concatenation, addition, etc.) input features and form new embeddings .

**Decoder**. We can freely choose from the multi-scale features computed during the *process* stage as inputs to the decoder module.

Empirically, we use the features on the original graph for prediction in all experiments. For node level tasks, we apply a last GNN layer on the original graph to get logits for every node. For graph level tasks, we first use global pooling functions to aggregate features. We can use common global pooling methods like SUM, MEAN, MAX or their combination. After global pool, we use MLP layer(s) to generate the prediction.

---

**Algorithm 2** Strided EdgePool

---

**Input:** graph $\mathcal{G} = (\mathcal{V}, \mathcal{E})$, edge scores $\mathbf{w}$, node pooling ratio $\eta_v$, maximum cluster sizes $\tau_c$.
**Output:** pooled graph $\tilde{\mathcal{G}} = (\tilde{\mathcal{V}}, \tilde{\mathcal{E}})$ and inter graph $\hat{\mathcal{G}} = (\hat{\mathcal{V}}, \hat{\mathcal{E}})$
INITIALIZE DISJOINT SET($\mathcal{G}$)
$remains \leftarrow N^v$                $\triangleright$ $N^v$ is the number of nodes in graph $\mathcal{G}$
$\bar{\mathcal{E}} \leftarrow$ Sort the edges $\mathcal{E}$ according to the edge scores $\mathbf{w}$ decreasingly.
**for** $e \in \bar{\mathcal{E}}$ **do**
     $x, y \leftarrow$ the two endpoints of the edge $e$
     $rx, sx \leftarrow$ FIND INDEX AND SIZE($x$)
     $ry, sy \leftarrow$ FIND INDEX AND SIZE($y$)
     **if** $rx \neq ry$ and $(sx + sy \leq \tau_c)$ **then**
         MERGE($x, y$)
         $remains \leftarrow remains - 1$
         **if** $remains \leq N^v * \eta_v$ **then**
             **break**
         **end if**
     **end if**
**end for**
$\tilde{\mathcal{V}}, \tilde{\mathcal{E}}, \hat{\mathcal{V}}, \hat{\mathcal{E}} \leftarrow \{\}, \{\}, \{\}, \{\}$
create empty mapping $n$ from cluster index to nodes
**for** $v \in \mathcal{V}$ **do**
     **if** FIND INDEX($v$) $= v$ **then**
         create new node $\tilde{v}$
         $n[v] = \tilde{v}$
         $\tilde{\mathcal{V}} \leftarrow \tilde{\mathcal{V}} \cup \{\tilde{v}\}$
     **end if**
**end for**
**for** $e \in \mathcal{E}$ **do**
     $x, y \leftarrow$ the two endpoints of the edge $e$
     $\tilde{x} \leftarrow n[$FIND INDEX($x$)$]$
     $\tilde{y} \leftarrow n[$FIND INDEX($y$)$]$
     $\tilde{\mathcal{E}} \leftarrow \tilde{\mathcal{E}} \cup \{(\tilde{x}, \tilde{y})\}$
**end for**
**for** $v \in \mathcal{V}$ **do**
     $\tilde{v} \leftarrow n[$FIND INDEX($v$)$]$
     $\hat{\mathcal{E}} \leftarrow \hat{\mathcal{E}} \cup \{(v, \tilde{v})\}$
**end for**
$\hat{\mathcal{V}} \leftarrow \mathcal{V} \cup \tilde{\mathcal{V}}$

---

## C.4 Architecture Variants

We can replace some GN blocks within `Mee` layers as an Identity block to reduce the time complexity. We call the height $j$ is reserved if the intra GN block of height $j$ is not replaced by an Identity block. We prefer to reserve a interval of consecutive heights for the `Mee` layers. (The inter GN blocks between these heights are remained unchanged while others are replaced as identities) By varying the heights reserved in each `Mee` layers, we can create a large amount of variants of `MeGraph` model including U-Shaped, Bridge-Shaped and Staircase-Shaped.

**U-Shaped**. This variant is similar to Graph U-Net (Gao & Ji, 2019). In this U-Shaped variant, the relationship between the number of layers $n$ and height $h$ is $n = 2h + 1$, and there is only one GN block in each layer. We keep the GN block at height $j = i$ for each layer $i$ at fist half layers, and keep the GN block at height $j = n - i + 1$ for each layer $i$ at later half layers. In the middle layer, only the last height $j = h = (n - 1)/2$ has a GN block.

**Bridge-Shaped**. In this variant, all GN blocks are combined like a arch bridge. Describe in detail, in the first and last layers, there are GN blocks in each height. In other layers, there are GN blocks at height of 1 to $j$ (where $1 < j < h$).

**Staircase-Shaped**. There are four forms in this variant, and the number of layers $n$ is equal to the height $h$ in all forms. The first from is like the 'downward' staircase. In each layer $i$ of this forms, there are GN blocks at height of $j$ to $h$ (where $j = i$). The second form is the inverted first form. In each layer $i$ of this second forms, there are GN blocks at height of 1 to $h - i + 1$ (where $j = i$). The last two forms are the mirror of the first and second forms.

# D   Implementation and Training Details

We use PyTorch (Paszke et al., 2019) and Deep Graph Library (DGL) (Wang et al., 2019) to implement our method.

We implement S-EdgePool using DGL, extending from the original implementation of EdgePool in the Pytorch Geometric library (PYG) (Fey & Lenssen, 2019). We did Constant optimization over the implementation to speed up the training and inference of the pooling. We further use Taichi-Lang (Hu et al., 2019) to speed up the dynamic node clustering process of S-EdgePool. The practical running time of `MeGraph` model with height $h > 1$ after optimization is about $2h$ times as the $h = 1$ baseline. This is still slower than the theoretical computational complexity due to the constant in the implementation and the difficulty of paralleling the sequential visitation of edges (according to their scores) in the EdgePool and S-EdgePool.

We run all our experiments on V100 GPUs and M40 GPUs. For training the neural networks, we use Adam (Kingma & Ba, 2015) as the optimizer. We report the hyper-parameters of the `Megraph` in Table 5.

For models using GFuN layer as the core GN block, we find it benefits from using layer norms (Ba et al., 2016). However, for models using GCN layer as the core GN block, we find it performs best when using batch norms (Ioffe & Szegedy, 2015).

The code will be made public, along with the configuration of hyper-parameters to reproduce our experiments.

# E   Additional Experiment Results

## E.1   TU Datasets

TU DATASETS consists of over 120 datasets of varying sizes from a wide range of applications. We choose 10 datasets, 5 of which are molecule datasets (MUTAG, NCI1, PROTEINS, D&D and ENZYMES) and the other 5 are social networks (IMDB-B, IMDB-M, REDDIT-BINARY, REDDIT-MULTI-5K and REDDIT-MULTI-12K). They are all graph classification tasks. For more details of each dataset, please refer to the original work (Morris et al., 2020).

Table 5: Hyper-parameters of the standard version of `MeGraph` for each dataset. It is worth noting that the total number of GNN layers is equals to one plus the number of `Mee` layers as $n + 1$.

| Hyper-parameters | Synthetic Datasets | Graph Theory Benchmark | GNN Benchmark | OGB Benchmark | TU Datasets |
|---|---|---|---|---|---|
| Repeated Runs | 10 | 5 | 4 | 5 | 1 for each fold |
| Epochs per run | 200 for BA* 500 for Tree* | 300 (200 for MCC) | 200 (100 for MNIST, CIFRA10) | 100 | 100 (200 for ENZYMES) |
| Learning rate | 0.002 | 0.002 (0.005 for MCC) | 0.001 | 0.001 | 0.002 |
| Weight decay | 0.0005 | 0.0005 | 0 | 0.0005 | 0.0005 |
| Node hidden dim | 64 | 128 | 144 | 300 | 128 |
| Edge hidden dim (for GFuN) | 64 | 128 | 144 | 300 | 128 |
| Num `Mee` layers $n$ | - | - | 3 | 4 | 2 |
| Height $h$ | - | - | 5 | 5 | 3 or 5 |
| Batch size | 32 | 32 | 128 | 32 | 128 |
| Input embedding | False | True | True | True | True |
| Global pooling | Mean | Mean Max | Mean | Mean | Mean Max Sum |
| Dataset split (train:val:test) | 8:1:1 | 8:1:1 | Original split | Original split | 10-fold cross validation |

Our `Megraph` uses the same network structure and hyper-parameters for the same type of dataset. As shown in Table 6, our `Megraph` achieves about 1% absolute gain than the $h$=1 Baselines.

Table 6: Tu Dataset Results Part 1. † means the results taken from Chen et al. (2019) (*: The result of GCN on ENZYMES is 100 epoch).

| Model | MUTAG ↑ | NCI1 ↑ | PROTEINS ↑ | D&D ↑ | ENZYMES ↑ | Average |
|---|---|---|---|---|---|---|
| GCN[†] | 87.20 ±5.11 | 83.65 ±1.69 | 75.65 ±3.24 | 79.12 ±3.07 | 66.50 ±6.91* | 78.42 |
| GIN[†] | 89.40 ±5.60 | 82.70 ±1.70 | 76.20 ±2.80 | - | - | - |
| GCN | 92.46 ±6.55 | 82.55 ±0.99 | 77.82 ±4.52 | 80.56 ±2.40 | 74.17 ±5.59 | 81.51 |
| MeGraph ($h$=1) | 93.01 ±6.83 | 82.53 ±1.89 | 81.32 ±4.08 | 81.32 ±3.17 | 74.83 ±3.20 | 82.60 |
| MeGraph | 93.07 ±6.71 | 83.99 ±0.98 | 81.41 ±3.10 | 81.24 ±2.39 | 75.17 ±4.86 | 82.98 |
| MeGraph_best | 94.12 ±5.02 | 84.40 ±1.11 | 81.68 ±3.40 | 82.00 ±2.86 | 75.17 ±4.86 | 83.47 |

| Model | IMDB-B ↑ | IMDB-M ↑ | RE-B ↑ | RE-M5K ↑ | RE-M12K ↑ | Average |
|---|---|---|---|---|---|---|
| GCN | 76.00 ±3.44 | 50.33 ±1.89 | 91.15 ±1.63 | 56.47 ±1.54 | 48.71 ±0.88 | 64.53 |
| MeGraph ($h$=1) | 68.60 ±3.53 | 51.33 ±2.23 | 93.10 ±1.16 | 57.47 ±2.31 | 51.56 ±1.06 | 64.41 |
| MeGraph | 72.40 ±2.80 | 51.27 ±2.71 | 93.75 ±1.25 | 57.69 ±2.22 | 52.03 ±0.86 | 65.43 |
| MeGraph_best | 74.30 ±2.97 | 52.00 ±2.49 | 93.75 ±1.25 | 58.45 ±2.22 | 52.13 ±1.01 | 66.13 |

Table 7: Comparison between GCN and GFuN on GNN benchmark.

| Model | ZINC ↓ | AQSOL ↓ | MNIST ↑ | CIFAR10 ↑ | PATTERN ↑ | CLUSTER ↑ |
|-------|--------|---------|---------|-----------|-----------|-----------|
| GCN | 0.426 ±0.015 | 1.397 ±0.029 | 90.140 ±0.140 | 51.050 ±0.390 | 84.672 ±0.054 | 47.541 ±0.940 |
| GFuN | 0.364 ±0.003 | 1.386 ±0.024 | 95.560 ±0.190 | 61.060 ±0.500 | 84.845 ±0.021 | 58.178 ±0.079 |

Table 8: Comparison between GCN and GFuN on OGB-G.

| Model | molhiv ↑ | molbace ↑ | molbbbp ↑ | molclintox ↑ | molsider ↑ |
|-------|----------|-----------|-----------|--------------|------------|
| GCN | 75.40 ±1.29 | 76.01 ±3.31 | 67.35 ±0.96 | 89.62 ±2.27 | 58.08 ±0.78 |
| GFuN | 78.54 ±1.14 | 71.77 ±2.15 | 67.56 ±1.11 | 89.77 ±3.48 | 58.28 ±0.51 |

| Model | moltox21 ↑ | moltoxcast ↑ | molesol ↓ | molfreesolv ↓ | mollipo ↓ |
|-------|------------|--------------|-----------|---------------|-----------|
| GCN | 75.11 ±0.41 | 64.13 ±0.52 | 1.141 ±0.02 | 2.407 ±0.15 | 0.788 ±0.01 |
| GFuN | 75.89 ±0.45 | 64.49 ±0.46 | 1.079 ±0.02 | 2.017 ±0.08 | 0.768 ±0.00 |

## E.2 GFuN

We show our GFuN results on real-world datasets compared to our reproduced GCN in Table 7, 8 and 9. Both GCN and GFuN have the same hyper-parameters except the batch norm for GCN and layer norm for GFuN as stated in Appendix D.

## E.3 SYNTHETIC DATASETS

Figure 6 shows the influence of the height $h$ and the number of `Mee` layers $n$ for `MeGraph` model on the BAShape and BACommunity datasets. The trend on these easier datasets is similar to that on TreeCycle and TreeGrid but less significant.

## E.4 GRAPH THEORY DATASET

We provide a list of tables (from Table 12 to 22) showing the individual results of Table 1 for each possible graph generation method. Each table contains a list of variants of models and 5 tasks. Some graph generation methods and task combinations are trivial so we filter them out.

Table 9: Comparison between GCN and GFuN on Tu Dataset.

| Model | MUTAG ↑ | NCI1 ↑ | PROTEINS ↑ | D&D ↑ | ENZYMES ↑ | Average |
|-------|---------|--------|------------|-------|-----------|---------|
| GCN | 92.46 ±6.55 | 82.55 ±0.99 | 77.82 ±4.52 | 80.56 ±2.40 | 74.17 ±5.59 | 81.51 |
| GFuN | 93.01 ±7.96 | 82.80 ±1.30 | 80.60 ±3.83 | 82.43 ±2.60 | 73.00 ±5.31 | 82.37 |

| Model | IMDB-B ↑ | IMDB-M ↑ | RE-B ↑ | RE-M5K ↑ | RE-M12K ↑ | Average |
|-------|----------|----------|--------|----------|-----------|---------|
| GCN | 76.00 ±3.44 | 50.33 ±1.89 | 91.15 ±1.63 | 56.47 ±1.54 | 48.71 ±0.88 | 64.53 |
| GFuN | 68.90 ±3.42 | 51.27 ±3.22 | 92.25 ±1.12 | 57.53 ±1.31 | 51.54 ±1.19 | 64.30 |

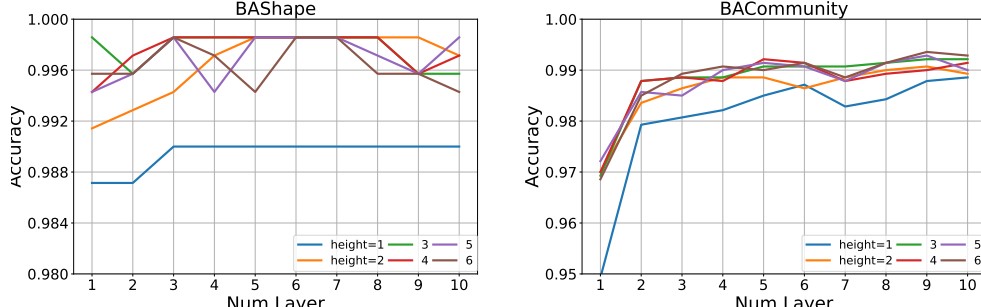

Figure 6: Node Classification accuracy for `MeGraph` model on BAShape (left) and BACommunity (right) datasets, varying the height $h$ and the number of `Mee` layers $n$. A clear gap can be observed between heights 1 and $\geq 2$. The concrete number of accuracy can be found in Table 11.

Table 10: Node Classification accuracy for `MeGraph` model on TreeCycle (above) and TreeGrid (below).

| layer \ height | 1 | 2 | 3 | 4 | 5 | 6 |
|---|---|---|---|---|---|---|
| 1 | 61.48 ±6.04 | 76.59 ±4.41 | 91.48 ±2.70 | 98.52 ±1.69 | 97.95 ±2.32 | 98.52 ±1.35 |
| 2 | 67.27 ±6.91 | 81.59 ±4.03 | 97.39 ±1.25 | 98.98 ±0.94 | 98.75 ±1.29 | 98.86 ±1.14 |
| 3 | 74.43 ±3.60 | 90.80 ±2.61 | 98.64 ±1.11 | 99.09 ±1.11 | 98.75 ±1.56 | 99.09 ±0.85 |
| 4 | 79.55 ±4.34 | 93.41 ±2.82 | 99.20 ±0.73 | 99.20 ±0.89 | 99.66 ±0.73 | 99.20 ±1.69 |
| 5 | 82.73 ±4.06 | 93.41 ±1.89 | 99.43 ±1.05 | 99.20 ±1.35 | 99.32 ±1.16 | 99.32 ±0.56 |
| 6 | 83.18 ±3.51 | 94.09 ±2.02 | 99.43 ±0.76 | 99.09 ±0.85 | 99.20 ±1.69 | 99.20 ±0.89 |
| 7 | 84.43 ±3.74 | 94.43 ±2.24 | 99.89 ±0.34 | 99.20 ±1.02 | 99.20 ±0.89 | 99.66 ±0.73 |
| 8 | 84.20 ±3.82 | 94.20 ±2.00 | 98.98 ±1.19 | 99.32 ±0.75 | 99.66 ±0.52 | 99.20 ±0.73 |
| 9 | 84.43 ±3.87 | 94.20 ±2.06 | 99.77 ±0.45 | 99.20 ±1.02 | 98.98 ±1.07 | 99.32 ±0.75 |
| 10 | 84.77 ±3.98 | 94.43 ±2.18 | 99.32 ±0.75 | 98.86 ±1.14 | 99.09 ±1.67 | 99.66 ±0.52 |

| layer \ height | 1 | 2 | 3 | 4 | 5 | 6 |
|---|---|---|---|---|---|---|
| 1 | 79.11 ±3.07 | 91.13 ±2.01 | 96.85 ±1.11 | 97.18 ±1.31 | 97.10 ±1.45 | 97.42 ±1.34 |
| 2 | 89.68 ±1.76 | 93.55 ±1.53 | 98.31 ±0.76 | 97.82 ±1.14 | 97.42 ±1.24 | 97.98 ±0.74 |
| 3 | 90.81 ±1.36 | 96.13 ±1.48 | 97.66 ±1.22 | 98.23 ±0.94 | 98.87 ±0.82 | 98.39 ±0.62 |
| 4 | 91.53 ±1.04 | 96.69 ±1.05 | 98.06 ±1.03 | 98.55 ±1.01 | 98.63 ±1.08 | 97.98 ±1.15 |
| 5 | 93.95 ±1.58 | 96.13 ±1.76 | 98.47 ±1.17 | 98.47 ±0.92 | 98.31 ±0.84 | 97.90 ±0.65 |
| 6 | 94.35 ±1.25 | 96.69 ±1.46 | 98.06 ±1.03 | 98.31 ±1.05 | 98.15 ±1.20 | 98.39 ±1.20 |
| 7 | 94.76 ±1.10 | 97.02 ±1.44 | 98.47 ±0.84 | 98.47 ±1.05 | 98.71 ±0.74 | 98.87 ±0.90 |
| 8 | 95.08 ±0.76 | 97.02 ±1.20 | 98.55 ±1.24 | 98.87 ±0.82 | 98.47 ±0.92 | 98.71 ±1.15 |
| 9 | 94.68 ±1.09 | 96.94 ±1.19 | 98.47 ±0.43 | 98.15 ±1.20 | 98.15 ±0.89 | 98.39 ±1.08 |
| 10 | 94.84 ±1.21 | 96.77 ±1.20 | 98.47 ±0.92 | 97.98 ±1.50 | 98.15 ±1.02 | 98.23 ±1.19 |

Table 11: Node Classification accuracy for `MeGraph` model on BAShape (above) and BACommunity (below).

| layer \ height | 1 | 2 | 3 | 4 | 5 | 6 |
|---|---|---|---|---|---|---|
| 1 | 98.71 ±1.00 | 99.14 ±1.14 | 99.86 ±0.43 | 99.43 ±0.70 | 99.43 ±0.95 | 99.57 ±0.91 |
| 2 | 98.71 ±1.00 | 99.29 ±0.96 | 99.57 ±0.91 | 99.71 ±0.57 | 99.57 ±0.91 | 99.57 ±0.91 |
| 3 | 99.00 ±0.91 | 99.43 ±0.95 | 99.86 ±0.43 | 99.86 ±0.43 | 99.86 ±0.43 | 99.86 ±0.43 |
| 4 | 99.00 ±0.91 | 99.71 ±0.57 | 99.86 ±0.43 | 99.86 ±0.43 | 99.43 ±0.95 | 99.71 ±0.57 |
| 5 | 99.00 ±0.91 | 99.86 ±0.43 | 99.86 ±0.43 | 99.86 ±0.43 | 99.86 ±0.43 | 99.43 ±0.95 |
| 6 | 99.00 ±0.91 | 99.86 ±0.43 | 99.86 ±0.43 | 99.86 ±0.43 | 99.86 ±0.43 | 99.86 ±0.43 |
| 7 | 99.00 ±0.91 | 99.86 ±0.43 | 99.86 ±0.43 | 99.86 ±0.43 | 99.86 ±0.43 | 99.86 ±0.43 |
| 8 | 99.00 ±0.91 | 99.86 ±0.43 | 99.86 ±0.43 | 99.86 ±0.43 | 99.71 ±0.57 | 99.57 ±0.65 |
| 9 | 99.00 ±0.91 | 99.86 ±0.43 | 99.57 ±0.91 | 99.57 ±0.91 | 99.57 ±0.91 | 99.57 ±0.91 |
| 10 | 99.00 ±0.91 | 99.71 ±0.57 | 99.57 ±0.91 | 99.71 ±0.57 | 99.86 ±0.43 | 99.43 ±0.95 |

| layer \ height | 1 | 2 | 3 | 4 | 5 | 6 |
|---|---|---|---|---|---|---|
| 1 | 94.93 ±1.30 | 97.00 ±1.80 | 96.93 ±1.60 | 97.00 ±1.88 | 97.21 ±1.70 | 96.86 ±1.67 |
| 2 | 97.93 ±0.87 | 98.36 ±0.72 | 98.79 ±0.46 | 98.79 ±0.46 | 98.57 ±0.55 | 98.50 ±1.03 |
| 3 | 98.07 ±0.91 | 98.64 ±0.87 | 98.86 ±0.91 | 98.86 ±0.80 | 98.50 ±0.98 | 98.93 ±0.80 |
| 4 | 98.21 ±0.97 | 98.86 ±0.65 | 98.86 ±0.80 | 98.79 ±0.64 | 99.00 ±0.73 | 99.07 ±0.64 |
| 5 | 98.50 ±0.87 | 98.86 ±0.91 | 99.07 ±0.64 | 99.21 ±0.67 | 99.14 ±0.70 | 99.00 ±0.73 |
| 6 | 98.71 ±0.83 | 98.64 ±0.87 | 99.07 ±0.64 | 99.14 ±0.70 | 99.07 ±0.85 | 99.14 ±0.70 |
| 7 | 98.29 ±0.91 | 98.86 ±0.65 | 99.07 ±0.56 | 98.79 ±0.56 | 98.79 ±0.72 | 98.86 ±0.57 |
| 8 | 98.43 ±0.77 | 99.00 ±0.47 | 99.14 ±0.53 | 98.93 ±0.58 | 99.14 ±0.29 | 99.14 ±0.43 |
| 9 | 98.79 ±0.79 | 99.07 ±0.56 | 99.21 ±0.50 | 99.00 ±0.73 | 99.29 ±0.45 | 99.36 ±0.50 |
| 10 | 98.86 ±0.73 | 98.93 ±0.80 | 99.21 ±0.87 | 99.14 ±0.70 | 99.00 ±0.73 | 99.29 ±0.64 |

Table 12: Graph Theory Benchmark results on Grid graphs, all results are obtained using our codebase.

| Category | Model | $SP_{sssd}$ | MCC | Diameter | $SP_{ss}$ | ECC |
|---|---|---|---|---|---|---|
| Baselines ($h=1$) | $n=1$ | 6.60 ±0.541 | 1.50 ±0.050 | 22.49 ±1.36 | 26.74 ±0.347 | 20.99 ±0.232 |
| | $n=5$ | 4.18 ±0.737 | 1.29 ±0.124 | 5.04 ±1.26 | 15.54 ±0.155 | 20.32 ±0.326 |
| | $n=10$ | 3.70 ±0.422 | 1.33 ±0.100 | 0.737 ±0.116 | 7.24 ±0.243 | 20.32 ±0.422 |
| MeGraph EdgePool ($h=5, \tau_c=2$) | $n=9$ (U-Shaped) | 2.12 ±1.07 | 2.04 ±0.206 | 2.14 ±0.991 | 2.01 ±0.212 | 19.39 ±0.996 |
| | $n=1$ | 1.19 ±0.486 | 1.24 ±0.154 | 6.78 ±1.95 | 5.34 ±0.265 | 18.00 ±0.910 |
| | $n=5$ | 0.738 ±0.322 | 1.11 ±0.043 | 0.616 ±0.310 | 0.617 ±0.099 | 13.3 ±3.31 |
| MeGraph S-EdgePool Ablation ($h=5, n=5$) | $\tau_c=3$ | 0.361 ±0.182 | 1.24 ±0.113 | 0.382 ±0.120 | 0.442 ±0.130 | 0.918 ±0.220 |
| | $\eta_v=0.3$ | 4.77 ±2.50 | 1.33 ±0.161 | 0.349 ±0.074 | 5.40 ±0.954 | 3.59 ±0.354 |
| | $\eta_v=0.3, \tau_c=4$ | 0.745 ±0.316 | 1.35 ±0.168 | 0.385 ±0.180 | 0.552 ±0.113 | 0.622 ±0.100 |
| | $\eta_v=0.5, \tau_c=4$ | 1.61 ±0.394 | 1.28 ±0.138 | 0.458 ±0.220 | 1.71 ±0.535 | 1.48 ±0.283 |
| | $\eta_v=0.3, \tau_c=4$ (X-Pool) | 1.03 ±0.365 | 1.50 ±0.142 | 0.626 ±0.216 | 1.70 ±0.185 | 3.44 ±0.991 |

Table 13: Graph Theory Benchmark results on Tree graphs. All results are obtained using our codebase.

| Category | Model | $SP_{sssd}$ | MCC | Diameter | $SP_{ss}$ | ECC |
|---|---|---|---|---|---|---|
| Baselines ($h=1$) | $n=1$ | 5.21 ±0.209 | 1.28 ±0.050 | 3.77 ±1.22 | 17.16 ±0.168 | 24.63 ±0.427 |
| | $n=5$ | 3.34 ±0.375 | 0.405 ±0.089 | 0.504 ±0.109 | 7.66 ±0.325 | 18.11 ±1.85 |
| | $n=10$ | 3.16 ±0.252 | 0.338 ±0.046 | 0.100 ±0.059 | 2.28 ±0.209 | 14.93 ±0.800 |
| MeGraph EdgePool ($h=5, \tau_c=2$) | $n=9$ (U-Shaped) | 3.73 ±1.01 | 1.30 ±0.092 | 1.36 ±0.623 | 4.21 ±0.440 | 25.53 ±5.43 |
| | $n=1$ | 1.62 ±0.314 | 0.846 ±0.071 | 0.725 ±0.249 | 6.99 ±0.610 | 12.27 ±0.843 |
| | $n=5$ | 0.83 ±0.667 | 0.490 ±0.118 | 0.084 ±0.030 | 1.27 ±0.442 | 2.87 ±0.420 |
| MeGraph S-EdgePool Ablation ($h=5, n=5$) | $\tau_c=3$ | 0.599 ±0.200 | 0.483 ±0.081 | 0.075 ±0.012 | 0.497 ±0.121 | 0.429 ±0.105 |
| | $\eta_v=0.3$ | 0.868 ±0.230 | 0.413 ±0.054 | 0.142 ±0.047 | 0.789 ±0.092 | 0.534 ±0.074 |
| | $\eta_v=0.3, \tau_c=4$ | 0.615 ±0.209 | 0.418 ±0.024 | 0.081 ±0.017 | 0.440 ±0.106 | 0.436 ±0.097 |
| | $\eta_v=0.5, \tau_c=4$ | 1.06 ±0.327 | 0.424 ±0.042 | 0.214 ±0.018 | 1.20 ±0.128 | 2.03 ±0.507 |
| | $\eta_v=0.3, \tau_c=4$ (X-Pool) | 0.666 ±0.118 | 0.596 ±0.067 | 0.182 ±0.057 | 1.22 ±0.281 | 1.11 ±0.122 |

Table 14: Graph Theory Benchmark results on Ladder graphs, all results are obtained using our codebase.

| Category | Model | $SP_{sssd}$ | MCC | Diameter | $SP_{ss}$ | ECC |
|---|---|---|---|---|---|---|
| Baselines (h=1) | $n$=1 | 5.06 ±0.330 | 1.73 ±0.249 | 1.17 ±0.149 | 13.20 ±0.126 | 20.10 ±0.583 |
| | $n$=5 | 0.692 ±0.204 | 0.734 ±0.106 | 1.39 ±0.078 | 5.02 ±0.876 | 19.81 ±0.669 |
| | $n$=10 | 0.257 ±0.078 | 0.691 ±0.119 | 1.55 ±0.069 | 1.60 ±0.194 | 20.40 ±0.995 |
| MeGraph EdgePool ($h$=5, $\tau_c$=2) | $n$=9 (U-Shaped) | 1.03 ±0.177 | 1.54 ±0.242 | 0.529 ±0.265 | 1.07 ±0.075 | 17.95 ±6.49 |
| | $n$=1 | 0.662 ±0.165 | 0.866 ±0.071 | 1.57 ±0.992 | 2.18 ±0.181 | 6.61 ±1.32 |
| | $n$=5 | 0.251 ±0.108 | 0.753 ±0.091 | 0.175 ±0.169 | 0.321 ±0.058 | 1.18 ±0.746 |
| MeGraph S-EdgePool Ablation ($h$=5, $n$=5) | $\tau_c$=3 | 0.296 ±0.070 | 0.754 ±0.086 | 0.226 ±0.069 | 0.228 ±0.021 | 0.285 ±0.069 |
| | $\eta_v$=0.3 | 0.507 ±0.204 | 0.768 ±0.050 | 0.156 ±0.053 | 0.969 ±0.148 | 0.787 ±0.059 |
| | $\eta_v$=0.3, $\tau_c$=4 | 0.297 ±0.113 | 0.712 ±0.059 | 0.095 ±0.046 | 0.180 ±0.026 | 0.225 ±0.043 |
| | $\eta_v$=0.5, $\tau_c$=4 | 0.375 ±0.196 | 0.656 ±0.064 | 0.058 ±0.019 | 0.612 ±0.191 | 0.464 ±0.121 |
| | $\eta_v$=0.3, $\tau_c$=4 (X-Pool) | 0.442 ±0.108 | 0.742 ±0.047 | 0.158 ±0.074 | 0.710 ±0.076 | 0.765 ±0.089 |

Table 15: Graph Theory Benchmark results on Line graphs, all results are obtained using our codebase.

| Category | Model | $SP_{sssd}$ | MCC | Diameter | $SP_{ss}$ | ECC |
|---|---|---|---|---|---|---|
| Baselines (h=1) | $n$=1 | 30.37 ±1.41 | 0.458 ±0.035 | 21.49 ±8.84 | 68.99 ±0.247 | 75.46 ±1.86 |
| | $n$=5 | 10.55 ±2.40 | 0.019 ±0.004 | 9.97 ±10.85 | 46.39 ±3.09 | 78.49 ±4.38 |
| | $n$=10 | 3.29 ±0.813 | 0.012 ±0.003 | 10.18 ±10.59 | 35.07 ±2.71 | 77.23 ±3.42 |
| MeGraph EdgePool ($h$=5, $\tau_c$=2) | $n$=9 (U-Shaped) | 1.95 ±1.11 | 0.355 ±0.080 | 5.79 ±2.22 | 2.68 ±1.12 | 74.39 ±14.30 |
| | $n$=1 | 1.45 ±0.598 | 0.056 ±0.014 | 7.62 ±4.43 | 10.13 ±2.33 | 45.19 ±8.64 |
| | $n$=5 | 0.536 ±0.149 | 0.016 ±0.007 | 0.611 ±0.238 | 1.06 ±0.341 | 14.12 ±13.82 |
| MeGraph S-EdgePool Ablation ($h$=5, $n$=5) | $\tau_c$=3 | 0.349 ±0.206 | 0.013 ±0.003 | 0.724 ±0.479 | 0.339 ±0.102 | 1.15 ±0.267 |
| | $\eta_v$=0.3 | 3.65 ±2.13 | 0.017 ±0.005 | 1.75 ±1.63 | 13.99 ±2.09 | 7.45 ±0.989 |
| | $\eta_v$=0.3, $\tau_c$=4 | 0.283 ±0.072 | 0.019 ±0.006 | 0.584 ±0.337 | 0.515 ±0.044 | 1.27 ±1.08 |
| | $\eta_v$=0.5, $\tau_c$=4 | 1.81 ±0.121 | 0.022 ±0.006 | 0.711 ±0.213 | 2.64 ±0.047 | 3.77 ±0.763 |
| | $\eta_v$=0.3, $\tau_c$=4 (X-Pool) | 1.06 ±0.510 | 0.101 ±0.016 | 0.767 ±0.522 | 2.29 ±0.472 | 3.89 ±1.02 |

Table 16: Graph Theory Benchmark results on Caterpillar graphs, all results are obtained using our codebase.

| Category | Model | $SP_{sssd}$ | MCC | Diameter | $SP_{ss}$ | ECC |
|---|---|---|---|---|---|---|
| Baselines (h=1) | $n$=1 | 24.24 ±1.57 | 1.25 ±0.082 | 28.62 ±2.55 | 19.08 ±0.208 | 35.32 ±0.462 |
| | $n$=5 | 8.32 ±2.10 | 0.561 ±0.070 | 4.59 ±0.346 | 9.62 ±0.357 | 37.01 ±1.48 |
| | $n$=10 | 6.40 ±0.652 | 0.630 ±0.127 | 5.06 ±0.499 | 4.06 ±0.297 | 37.87 ±3.22 |
| MeGraph EdgePool ($h$=5, $\tau_c$=2) | $n$=9 (U-Shaped) | 6.66 ±1.25 | 1.39 ±0.098 | 8.64 ±3.46 | 2.63 ±0.211 | 32.18 ±4.30 |
| | $n$=1 | 5.04 ±1.03 | 0.685 ±0.077 | 6.08 ±1.40 | 5.40 ±0.843 | 28.52 ±2.16 |
| | $n$=5 | 3.44 ±1.13 | 0.533 ±0.064 | 2.00 ±1.28 | 0.921 ±0.149 | 5.20 ±1.57 |
| MeGraph S-EdgePool Ablation ($h$=5, $n$=5) | $\tau_c$=3 | 2.47 ±0.529 | 0.607 ±0.081 | 0.591 ±0.172 | 0.574 ±0.073 | 1.21 ±0.148 |
| | $\eta_v$=0.3 | 3.61 ±1.36 | 0.582 ±0.052 | 0.578 ±0.231 | 1.69 ±0.572 | 1.95 ±0.322 |
| | $\eta_v$=0.3, $\tau_c$=4 | 1.59 ±0.444 | 0.535 ±0.091 | 0.317 ±0.104 | 0.474 ±0.170 | 1.32 ±0.272 |
| | $\eta_v$=0.5, $\tau_c$=4 | 2.00 ±0.648 | 0.514 ±0.040 | 1.10 ±0.288 | 0.986 ±0.130 | 2.11 ±0.766 |
| | $\eta_v$=0.3, $\tau_c$=4 (X-Pool) | 1.39 ±0.478 | 0.602 ±0.110 | 0.736 ±0.230 | 1.78 ±0.254 | 3.36 ±0.873 |

Table 17: Graph Theory Benchmark results on Lobster graphs, all results are obtained using our codebase.

| Category | Model | $SP_{sssd}$ | MCC | Diameter | $SP_{ss}$ | ECC |
|---|---|---|---|---|---|---|
| Baselines (h=1) | $n$=1 | 23.92 ±0.319 | 1.06 ±0.166 | 11.93 ±1.32 | 38.44 ±0.065 | 40.46 ±0.350 |
| | $n$=5 | 10.89 ±1.47 | 0.544 ±0.067 | 3.66 ±0.424 | 20.12 ±0.105 | 28.81 ±1.14 |
| | $n$=10 | 7.35 ±2.50 | 0.631 ±0.067 | 2.59 ±0.517 | 10.52 ±0.619 | 28.47 ±1.65 |
| MeGraph EdgePool ($h$=5, $\tau_c$=2) | $n$=9 (U-Shaped) | 6.20 ±1.13 | 1.56 ±0.217 | 5.65 ±1.17 | 7.06 ±0.671 | 29.07 ±2.59 |
| | $n$=1 | 6.00 ±1.82 | 0.785 ±0.062 | 4.35 ±1.51 | 13.75 ±0.675 | 30.49 ±2.18 |
| | $n$=5 | 1.93 ±0.861 | 0.543 ±0.073 | 1.07 ±0.114 | 2.05 ±0.393 | 11.39 ±5.43 |
| MeGraph S-EdgePool Ablation ($h$=5, $n$=5) | $\tau_c$=3 | 2.02 ±0.791 | 0.447 ±0.123 | 0.705 ±0.133 | 1.66 ±0.270 | 2.23 ±0.378 |
| | $\eta_v$=0.3 | 6.01 ±1.52 | 0.521 ±0.028 | 0.707 ±0.202 | 3.04 ±0.250 | 2.70 ±0.212 |
| | $\eta_v$=0.3, $\tau_c$=4 | 1.90 ±0.449 | 0.489 ±0.069 | 0.671 ±0.165 | 1.30 ±0.106 | 2.62 ±0.849 |
| | $\eta_v$=0.5, $\tau_c$=4 | 3.27 ±0.716 | 0.451 ±0.090 | 0.941 ±0.324 | 2.82 ±0.803 | 4.04 ±0.527 |
| | $\eta_v$=0.3, $\tau_c$=4 (X-Pool) | 2.67 ±0.486 | 0.494 ±0.109 | 1.01 ±0.194 | 2.79 ±0.343 | 4.16 ±0.886 |

Table 18: Graph Theory Benchmark results on Cycle graphs, all results are obtained using our codebase.

| Category | Model | $SP_{sssd}$ | MCC | Diameter | $SP_{ss}$ | ECC |
|---|---|---|---|---|---|---|
| Baselines (h=1) | $n$=1 | 18.75 ±0.066 | 0.534 ±0.022 | 22.35 ±0.149 | 24.07 ±0.009 | 21.47 ±0.060 |
| | $n$=5 | 3.39 ±0.304 | 0.027 ±0.001 | 25.11 ±0.325 | 12.44 ±1.05 | 21.81 ±0.102 |
| | $n$=10 | 0.352 ±0.060 | 0.011 ±0.003 | 26.54 ±1.16 | 8.65 ±1.02 | 24.09 ±0.360 |
| MeGraph EdgePool ($h$=5, $\tau_c$=2) | $n$=9 (U-Shaped) | 0.964 ±0.742 | 0.251 ±0.073 | 22.27 ±3.71 | 0.910 ±0.121 | 25.32 ±3.98 |
| | $n$=1 | 0.594 ±0.212 | 0.074 ±0.029 | 9.11 ±1.88 | 4.07 ±0.364 | 21.53 ±0.070 |
| | $n$=5 | 0.060 ±0.032 | 0.014 ±0.003 | 13.44 ±6.40 | 0.103 ±0.016 | 24.05 ±0.204 |
| MeGraph S-EdgePool Ablation ($h$=5, $n$=5) | $\tau_c$=3 | 0.066 ±0.036 | 0.015 ±0.006 | 0.241 ±0.049 | 0.090 ±0.037 | 0.342 ±0.186 |
| | $\eta_v$=0.3 | 2.45 ±0.873 | 0.015 ±0.001 | 0.709 ±0.226 | 8.36 ±0.261 | 0.488 ±0.267 |
| | $\eta_v$=0.3, $\tau_c$=4 | 0.060 ±0.030 | 0.019 ±0.003 | 0.312 ±0.236 | 0.226 ±0.050 | 0.562 ±0.209 |
| | $\eta_v$=0.5, $\tau_c$=4 | 0.451 ±0.203 | 0.014 ±0.004 | 0.252 ±0.124 | 1.05 ±0.524 | 4.30 ±1.90 |
| | $\eta_v$=0.3, $\tau_c$=4 (X-Pool) | 0.494 ±0.292 | 0.096 ±0.028 | 0.468 ±0.220 | 1.08 ±0.130 | 0.860 ±0.292 |

Table 19: Graph Theory Benchmark results on Pseudotree graphs, all results are obtained using our codebase.

| Category | Model | $SP_{sssd}$ | MCC | Diameter | $SP_{ss}$ | ECC |
|---|---|---|---|---|---|---|
| Baselines (h=1) | $n$=1 | 1.93 ±0.239 | 1.71 ±0.281 | 2.78 ±0.098 | 6.27 ±0.004 | 4.23 ±0.034 |
| | $n$=5 | 0.061 ±0.024 | 0.942 ±0.094 | 1.74 ±0.299 | 1.54 ±0.006 | 4.15 ±0.086 |
| | $n$=10 | 0.037 ±0.022 | 0.775 ±0.094 | 1.84 ±0.260 | 0.126 ±0.038 | 4.06 ±0.037 |
| MeGraph EdgePool ($h$=5, $\tau_c$=2) | $n$=9 (U-Shaped) | 1.12 ±0.229 | 2.55 ±0.148 | 2.53 ±0.425 | 1.26 ±0.129 | 4.14 ±0.135 |
| | $n$=1 | 0.404 ±0.096 | 1.75 ±0.133 | 1.50 ±0.494 | 2.25 ±0.280 | 3.97 ±0.270 |
| | $n$=5 | 0.141 ±0.022 | 0.999 ±0.054 | 1.16 ±0.069 | 0.148 ±0.034 | 3.12 ±0.202 |
| MeGraph S-EdgePool Ablation ($h$=5, $n$=5) | $\tau_c$=3 | 0.130 ±0.069 | 0.912 ±0.073 | 0.669 ±0.080 | 0.115 ±0.015 | 0.797 ±0.079 |
| | $\eta_v$=0.3 | 0.048 ±0.030 | 0.839 ±0.077 | 0.758 ±0.134 | 0.246 ±0.021 | 0.838 ±0.023 |
| | $\eta_v$=0.3, $\tau_c$=4 | 0.106 ±0.054 | 0.814 ±0.092 | 0.663 ±0.076 | 0.133 ±0.028 | 0.845 ±0.101 |
| | $\eta_v$=0.5, $\tau_c$=4 | 0.071 ±0.048 | 1.03 ±0.186 | 0.583 ±0.065 | 0.171 ±0.038 | 0.868 ±0.034 |
| | $\eta_v$=0.3, $\tau_c$=4 (X-Pool) | 0.564 ±0.155 | 0.966 ±0.172 | 0.977 ±0.054 | 0.611 ±0.065 | 1.10 ±0.036 |

Table 20: Graph Theory Benchmark results on Geo graphs, all results are obtained using our codebase.

| Category | Model | $SP_{sssd}$ | MCC | Diameter | $SP_{ss}$ | ECC |
|---|---|---|---|---|---|---|
| Baselines (h=1) | $n$=1 | 5.79 ±0.630 | 0.424 ±0.023 | 11.85 ±0.391 | 12.49 ±0.035 | 14.82 ±0.056 |
| | $n$=5 | 1.02 ±0.772 | 0.407 ±0.040 | 8.37 ±0.468 | 5.10 ±0.435 | 14.33 ±0.079 |
| | $n$=10 | 0.304 ±0.125 | 0.404 ±0.061 | 9.41 ±0.759 | 0.803 ±0.162 | 14.33 ±0.136 |
| MeGraph EdgePool ($h$=5, $\tau_c$=2) | $n$=9 (U-Shaped) | 2.99 ±0.373 | 0.549 ±0.114 | 7.66 ±1.61 | 3.38 ±0.557 | 12.67 ±0.699 |
| | $n$=1 | 1.60 ±0.880 | 0.347 ±0.033 | 10.17 ±2.04 | 4.87 ±0.777 | 11.91 ±0.451 |
| | $n$=5 | 0.232 ±0.061 | 0.273 ±0.018 | 2.70 ±0.288 | 0.575 ±0.127 | 6.92 ±2.36 |
| MeGraph S-EdgePool Ablation ($h$=5, $n$=5) | $\tau_c$=3 | 0.188 ±0.100 | 0.288 ±0.020 | 2.04 ±0.225 | 0.562 ±0.186 | 2.42 ±0.333 |
| | $\eta_v$=0.3 | 1.38 ±0.617 | 0.330 ±0.025 | 4.40 ±1.15 | 1.37 ±0.083 | 5.45 ±0.465 |
| | $\eta_v$=0.3, $\tau_c$=4 | 0.230 ±0.070 | 0.231 ±0.034 | 1.99 ±0.549 | 0.454 ±0.057 | 2.69 ±0.369 |
| | $\eta_v$=0.5, $\tau_c$=4 | 0.374 ±0.148 | 0.368 ±0.043 | 3.95 ±0.319 | 0.777 ±0.122 | 4.61 ±0.717 |
| | $\eta_v$=0.3, $\tau_c$=4 (X-Pool) | 1.04 ±0.502 | 0.362 ±0.031 | 2.32 ±0.440 | 2.37 ±0.260 | 5.08 ±0.737 |

Table 21: Graph Theory Benchmark results on BA graphs, all results are obtained using our codebase.

| Category | Model | $SP_{sssd}$ | MCC | Diameter | $SP_{ss}$ | ECC |
|---|---|---|---|---|---|---|
| Baselines (h=1) | $n$=1 | 0.004 ±0.001 | 2.81 ±0.142 | 0.092 ±0.021 | − | 0.128 ±0.006 |
| | $n$=5 | 0.007 ±0.002 | 3.65 ±0.660 | 0.098 ±0.014 | − | 0.091 ±0.011 |
| | $n$=10 | 0.011 ±0.006 | 3.72 ±0.376 | 0.122 ±0.038 | − | 0.080 ±0.004 |
| MeGraph EdgePool ($h$=5, $\tau_c$=2) | $n$=9 (U-shape) | 0.060 ±0.021 | 1.61 ±0.233 | 0.219 ±0.055 | − | 0.115 ±0.030 |
| | $n$=1 | 0.006 ±0.004 | 2.00 ±0.380 | 0.101 ±0.020 | − | 0.084 ±0.017 |
| | $n$=5 | 0.003 ±0.001 | 2.00 ±0.240 | 0.104 ±0.011 | − | 0.052 ±0.010 |
| MeGraph S-EdgePool Ablation ($h$=5, $n$=5) | $\tau_c$=3 | 0.007 ±0.003 | 1.77 ±0.403 | 0.089 ±0.008 | − | 0.126 ±0.027 |
| | $\eta_v$=0.3 | 0.013 ±0.004 | 1.67 ±0.333 | 0.084 ±0.008 | − | 0.086 ±0.005 |
| | $\eta_v$=0.3, $\tau_c$=4 | 0.011 ±0.005 | 1.42 ±0.252 | 0.073 ±0.015 | − | 0.163 ±0.007 |
| | $\eta_v$=0.5, $\tau_c$=4 | 0.008 ±0.004 | 1.71 ±0.403 | 0.074 ±0.009 | − | 0.156 ±0.021 |
| | $\eta_v$=0.3, $\tau_c$=4 (X-Pool) | 0.009 ±0.003 | 1.22 ±0.242 | 0.088 ±0.021 | − | 0.076 ±0.006 |

Table 22: Graph Theory Benchmark results on mixed, ER, Caveman and Star graphs, all results are obtained using our codebase.

| Category | Model | MCC | | | | ECC | |
|---|---|---|---|---|---|---|---|
| | | mix | ER | Caveman | Star | mix | ER |
| Baselines $(h=1)$ | $n=1$ | 3.46 ±0.211 | 2.91 ±0.206 | 0.015 ±0.004 | 0.144 ±0.031 | 0.316 ±0.003 | 0.346 ±0.006 |
| | $n=5$ | 3.29 ±0.261 | 3.35 ±0.205 | 0.014 ±0.003 | 0.078 ±0.021 | 0.228 ±0.008 | 0.289 ±0.008 |
| | $n=10$ | 3.51 ±0.323 | 3.53 ±0.375 | 0.018 ±0.006 | 0.065 ±0.005 | 0.212 ±0.008 | 0.414 ±0.102 |
| MeGraph EdgePool $(h=5, \tau_c=2)$ | $n=9$ (U-Shaped) | 1.63 ±0.078 | 1.02 ±0.128 | 0.091 ±0.026 | 0.125 ±0.032 | 0.200 ±0.013 | 0.272 ±0.008 |
| | $n=1$ | 1.25 ±0.167 | 0.749 ±0.058 | 0.018 ±0.005 | 0.135 ±0.055 | 0.150 ±0.011 | 0.320 ±0.071 |
| | $n=5$ | 1.11 ±0.143 | 0.723 ±0.073 | 0.017 ±0.005 | 0.052 ±0.017 | 0.125 ±0.010 | 0.345 ±0.064 |
| MeGraph S-EdgePool Ablation $(h=5, n=5)$ | $\tau_c=3$ | 1.07 ±0.034 | 0.714 ±0.039 | 0.017 ±0.002 | 0.072 ±0.016 | 0.137 ±0.013 | 0.232 ±0.035 |
| | $\eta_v=0.3$ | 0.908 ±0.153 | 0.627 ±0.090 | 0.026 ±0.007 | 0.125 ±0.026 | 0.128 ±0.014 | 0.248 ±0.012 |
| | $\eta_v=0.3, \tau_c=4$ | 1.10 ±0.085 | 0.709 ±0.092 | 0.019 ±0.004 | 0.073 ±0.012 | 0.129 ±0.009 | 0.224 ±0.053 |
| | $\eta_v=0.5, \tau_c=4$ | 1.12 ±0.219 | 0.722 ±0.128 | 0.026 ±0.008 | 0.058 ±0.010 | 0.147 ±0.017 | 0.219 ±0.042 |
| | $\eta_v=0.3, \tau_c=4$ (X-Pool) | 1.01 ±0.166 | 0.838 ±0.078 | 0.029 ±0.007 | 0.107 ±0.021 | 0.119 ±0.008 | 0.213 ±0.027 |

