# OpenReview forum: "MeGraph: Graph Representation Learning on Connected Multi-scale Graphs"
_ICLR.cc/2023/Conference — Submitted to ICLR 2023_

### Official Review · Reviewer_M2DM · 2022-10-22

**Confidence:** 4
**Correctness:** 3
**Technical Novelty And Significance:** 2
**Empirical Novelty And Significance:** 2
**Recommendation:** 5

**Clarity, Quality, Novelty And Reproducibility:**

The paper is clearly written, but the organization of the paper could be improved. The idea of the paper is incremental. The proposed framework works better than basic GNNs but further comparisons are necessary.

**Strength And Weaknesses:**

Strengths:
1. The idea of the paper is straightforward and easy to understand.
2. Experimental evaluation is performed on various tasks and datasets.

Weaknesses:
1. The proposed framework is incremental, which limits the technical contribution.
The three main components, the graph pooling, the GN layer, and the encode-decode architecture, are all adopted from existing methods.

2. The organization of the paper is poor and should be improved.
Graph pooling is well-known and widely used in GNNs. There is no need to introduce it in great details. It is suggested to put more details of the proposed method in the main body.

3. The expressiveness power of the proposed method needs further theoretical evaluations.
The paper aims to address the inability of a GNN node to observe nodes that are farther away than the number of layers n (so-called the under-reaching problem). This problem was first discussed in [1]. In this work, however, introducing large-scale graph representations to update node representations could aggravate over-squashing. Moreover, from the perspective of graph isomorphism, connecting nodes from different scales could harm the ability of GNN to distinguish non-isomorphic graphs. The authors should analyze how powerful the proposed framework is, in theory, to make sure it addresses the under-reaching problem.

4. Experimental study should be enhanced.
(1) The proposed method only compares with basic GNNs, such as GCN and GIN. More SOTA methods should be added as baselines. For instance, [2] and [3] also try to extend receptive fields by rewiring the input graph.
(2) The selection of each component in MeGraph could significantly influence the performance. The discussion on the selection of each component, such as the pooling method and the GN block, is insufficient.
(3) Since the GN block in the proposed method is realized as a GFuN layer, the authors should report the experimental results of GFuN to clarify the performance improvements that come from the proposed MeGraph.

[1] Uri Alon and Eran Yahav. On the bottleneck of graph neural networks and its practical implications. In International Conference on Learning Representations, 2021.
[2] Topping J, Di Giovanni F, Chamberlain B P, et al. Understanding over-squashing and bottlenecks on graphs via curvature. In International Conference on Learning Representations, 2022.
[3] Brüel-Gabrielsson R, Yurochkin M, Solomon J. Rewiring with positional encodings for graph neural networks. arXiv preprint arXiv:2201.12674, 2022.


**Summary Of The Paper:**

The paper proposes a mega graph structure and introduces it to a GNN named MeGraph. MeGraph aims to obtain better node representations by enabling repeated information exchange across multi-scale graphs.  Experimental results are provided to demonstrate the effectiveness of MeGraph.

**Summary Of The Review:**

Main concerns of the paper are: (1) The technical contribution of the paper is limited. (2) The expressiveness of the proposed model should be analysed theoretically. (3) Baseline models are weak. Comparisons with SOTA models are needed. (4) MeGraph intensely relies on the graph pooling and the GN layer, which are well-studied and have different options. The discussion on the selection of these components is insufficient.

Post-rebuttal: I appreciate the authors' efforts in providing theoretic proof and additional experimental results, which have partially addressed my concerns. However, my concerns on the limited technical contribution and the lack of extensive experiments for the universality of the framework still remain. I have slightly increased the rating.

---

> ### Author Response · Authors · 2022-11-18
> **Respond to Reviewer M2DM**
>
> Thanks for providing us with the references related to the under-reaching problem.
>
> > **1**. The three main components, the graph pooling, the GN layer, and the encode-decode architecture, are all adopted from existing methods.
>
> We would like to re-clarify our main contributions.
> 1. **We provide a novel perspective from a mega graph for multi-scale graph representation learning problems**. Note that **the structure of the mega graph** (i.e., the structure of multi-scale graphs and connections between the successive scales) **is not fixed but evolved along with the training of graph pooling and convolutions**.
> 2. **We utilize the hierarchical nature of the mega graph and design the novel Mee layer that achieves efficient cross-scale update**  (detailed in Sec 3.2).
> 3. **We propose S-EdgePool that addresses the original EdgePool's fixed pooling ratio issue using a novel cluster-finding algorithm** (detailed in Sec 3.3).
> 4. **MeGraph demonstrates dominated performance compared with the baselines in the graph theory benchmark**.
> In summary, our work is NOT a combination of existing GNN modules (including graph convs, and graph poolings), but rather a novel framework for multi-scale graph representation learning, which expertises in tasks that require considering long-range relationships. **Note that these modules are not trained separately (like first train graph pooling and then graph convs) but jointly together within the MeGraph model**.
>
> Please refer to our posted **General Message** about novelty for more details.
>
> > **2**. Graph pooling is well-known and widely used in GNNs. There is no need to introduce it in great details.
>
> **Answer**: We need to clarify a few reasons why we introduce graph pooling with more contexts in the preliminaries:
> 1. Our notations differ slightly from those used in the SRC framework (Grattarola et al., 2022).
> 2. Generalizing over the SRC framework, our formulation of graph pooling further introduces the inter graph $\hat{\mathcal{G}}$ and its features $\mathbf{X}^{\hat{\mathcal{G}}}$, which is essential for us to explain how to build the mega-graph using inter graphs.
> 3. We extend the EdgePool (Diehl et al., 2019) method to S-EdgePool with arbitrary pooling ratios, so we have to introduce the original EdgePool with details in advance.
>
> > **3a**. The expressiveness power of the proposed method needs further theoretical evaluations. (...) The authors should analyze how powerful the proposed framework is, in theory, to make sure it addresses the under-reaching problem.
>
> **Answer**: Thanks for the suggestion. We could briefly explain why MeGraph can address the under-reaching problem.
>
> MeGraph operates on a mega graph constructed by connecting graph pyramids. From the perspective of the mega graph, the distance between any two nodes from the original input graph is at most $2h$, where $h$ is the height of the mega graph, considering the back-and-forth message passing along the inter-edges. When we adopt a pooling method that can reduce a graph with half the size of the input graph, $h$ is at most of $O(\log(n))$, which could be sufficiently smaller than the original distance of the two nodes in the original input graph (which could be of $O(n)$ distance).
>
> In practice, we have created 55 Graph Theory Benchmarks, most of which suffer from the problem of under-reaching. The results show that MeGraph is extremely effective in these graph theory problems.
>
> > **3b**. In this work, however, introducing large-scale graph representations to update node representations could aggravate over-squashing.
>
> **Answer**:
> We agree that introducing large-scale graph representations might aggravate the over-squashing problem. However, this is off the point under the scope of the current work. We do not count how much information is over-squashed; instead, we should focus on the quality of the graph representation, i.e., the quality and representation power of the output embedding and how much performance gain it brings.

---

> > ### Author Response · Authors · 2022-11-18
> > **Respond to Reviewer M2DM Cont'd**
> >
> > > **3c**. Moreover, from the perspective of graph isomorphism, connecting nodes from different scales could harm the ability of GNN to distinguish non-isomorphic graphs.
> >
> > **Answer**: Thanks for discussing MeGraph from the point of graph isomorphism. The MeGraph structure supports the gating scheme when aggregating features from different scales. Therefore, it can theoretically learn to gate out the features from other scales and focus on the original graph only (degenerate to the vanilla GNNs). So, MeGraph should at least have as much ability to distinguish non-isomorphic graphs as GNNs.
> >
> > > **4a**. The proposed method only compares with basic GNNs, such as GCN and GIN. More SOTA methods should be added as baselines. For instance, [2] and [3] also try to extend receptive fields by rewiring the input graph.
> >
> > **Answer**: Please refer to our posted **General message** about comparing to other baselines.
> >
> > For [2] and [3], the graph rewiring and positional encoding methods are orthogonal to our framework. In both the encoder and decoder paragraphs of Sec 3.3 and Appendix C.3, we stated that we could adopt different positional encoding methods to further enhance the expressiveness. This statement also holds for graph rewiring methods as they are preprocessing steps before training.
> >
> > > **4b**. The selection of each component in MeGraph could significantly influence the performance. The discussion on the selection of each component, such as the pooling method and the GN block, is insufficient.
> >
> > **Answer**:
> > We would like to explain why we choose EdgePool and GFuN as the core components.
> >
> > Grattarola et al. (2022) propose a taxonomy of pooling operations into trainability, density, adaptability, and hierarchy. We choose EdgePool as our starting point as it is trainable, sparse, adaptable, hierarchy, and well preserves the graph's overall structure (**stated in the top paragraph of Page 6**). We further strengthen its adaptability by allowing varying the pooling ratio.
> >
> > GFuN is a realization of a full GN block (Battaglia et al., 2018), which covers a large set of variants of GNN layers. We choose GFuN because of its excellent flexibility and ability to handle graph, node, and edge features. It also shows performance gain over GCN in Tables 7, 8, and 9.
> >
> > We have revised Sec 3.3 accordingly.
> >
> > There exist more candidate components in the literature, while it is impractical to enumerate all possible choices and combinations.
> >
> > > **4c**. Since the GN block in the proposed method is realized as a GFuN layer, the authors should report the experimental results of GFuN to clarify the performance improvements that come from the proposed MeGraph.
> >
> > **Answer**: We did report the comparison between GCN and GFuN in Tables 7, 8, and 9 in Appendix E.2.

---

> > > ### Comment · Reviewer_M2DM · 2022-12-09
> > > **Thank you**
> > >
> > > Thank you for the response. However, some of my concerns are not well addressed and thus I will keep my rating.
> > >
> > > 1. Although the mega graph is a new concept, the components of MeGraph are adopted or modified from existing works, which limits the novelty of the proposed method.
> > > 2. The organization of the paper could be improved.
> > > 3. The claim "MeGraph should at least have as much ability to distinguish non-isomorphic graphs as GNNs” is incorrect. It is possible that a pair of GNN-distinguishable graphs gets the same representation by information exchange across multi-scale graphs that make it indistinguishable. The authors should provide rigorous theoretical analysis to support their claim.
> > > 4. The author didn’t provide sufficient experimental study (plugin to SOTA GNNs and ablation study) to demonstrate the effectiveness of their method. As the theoretical contribution of this paper is not strong, the experiments are expected to be extensive. In particular, an empirical validation on the selection of each component is expected. There are many different choices for each component. Showing only one option for each component is not good enough. I am not asking for an exhaustive enumeration, but it is necessary to at least show some other alternatives in order to validate the effectiveness of the framework, that is, to show that the framework is general and the improvement doesn't rely on a specific combination.

---

> > > > ### Author Response · Authors · 2022-12-11
> > > > **Follow-up (Part 1/2)**
> > > >
> > > > Thanks for your response, we provide a point-by-point response below.
> > > >
> > > > > 1. Although the mega graph is a new concept, the components of MeGraph are adopted or modified from existing works, which limits the novelty of the proposed method.
> > > >
> > > > Thanks for acknowledging the novelty of the mega graph. Therefore, the MeGraph architecture and the Mee Layer built upon the novel mega graph should also be novel. **We do NOT think the novelty of a new architechture should be lowered down just because it uses existing modules**.
> > > >
> > > > > 2. The organization of the paper could be improved.
> > > >
> > > > We have explained in the last response why we must introduce graph poolings in details.
> > > > We would like to, but we could not put more details in the main body due to the space limit. We would revise again with a better organization to include the points mentioned in the rebuttal.
> > > >
> > > > > 3. It is possible that a pair of GNN-distinguishable graphs gets the same representation by information exchange across multi-scale graphs that make it indistinguishable. The authors should provide rigorous theoretical analysis to support their claim.
> > > >
> > > > As explained in the last response: "The MeGraph structure supports the gating scheme when aggregating features from different scales," MeGraph can learn a gating function (within the XUPD function) that only reserves the features of the same scale while performing cross-scale information exchanging. In that case, there will be no information exchange across multi-scale graphs, and features other than those in the original scale will not be aggregated. Therefore, a pair of GNN-distinguishable graphs will also get different representations within MeGraph. We provide rigorous proof below and will include this in revision.
> > > >
> > > > **Proof**: The cross update function is $(X_j', \hat{X}_j', X_\{j+1\}') = XUPD(j, X_j, \hat{X_j}, X_\{j+1\})$. There is a residual function applied here, and we assume it is implemented as a gated residual: $X_j'' = \sigma(\alpha) X_j + \sigma(\beta) X_j'$, where $\sigma$ is the sigmoid function and $\alpha, \beta$ are learnable parameters. Theoremetically, it is possible that $\sigma(\alpha)=1$ and $\sigma(\beta)=0$ after training. In that case, $X_j'' = X_j$, which means $X_j$ is not changed over steps 2 and 3 of the Mee layer. Therefore, $X_0^i = GN_\{intra\}^\{i, 0\}(\mathcal{G},X_0^\{i-1\})$, this is equivalent to a simple GNN layer that $X^i = GNN_i(\mathcal{G}, X^\{i-1\})$ as $X_0^i$ is the features of the original graph and $GN_\{intra\}$ is a GNN layer. Therefore, MeGraph degenerates to ordinary GNN in this case, $MeGraph(\mathcal{G}) = GNN(\mathcal{G})$. For any pair of GNN-distinguishable graphs, where $GNN(\mathcal{G}_1) \neq GNN(\mathcal{G}_2)$, MeGraph can also distinguish as $MeGraph(\mathcal{G}_1) = GNN(\mathcal{G}_1) \neq GNN(\mathcal{G}_2) = MeGraph(\mathcal{G}_2)$. $\blacksquare$
> > > >
> > > > > 4. The author didn’t provide sufficient experimental study (plugin to SOTA GNNs and ablation study) to demonstrate the effectiveness of their method. There are many different choices for each component. Showing only one option for each component is not good enough.
> > > >
> > > > It is hard to define what are SOTA GNNs. Please note that GFuN, as a realization of the full GN block (Battaglia et al., 2018), has a highly configurable within-block structure and is therefore capable of expressing a variety of other architectures (Please see Sec 4.2 of Battaglia et al., 2018), like GCN, GIN, GAT, GatedGCN. Therefore, varying the within-block structure of GFuN is equivalent to plugin different GNN cores. GFuN is **NOT a single option** but a collection of methods when varying arguments. **We DO observe a consistent improvement for the different within-block structures of GFuN** and provide detailed results below (in the follow-up part 2/2).
> > > >
> > > > As for graph pooling, note that EdgePool and S-EdgePool are already different pooling methods and can have very different pooling outcomes. It can be regarded as **different poolings by varying the argument $\eta_v$ and $\tau_c$ of S-EdgePool**. As illustrated in the ablation category of Table 1, no matter which combination of arguments is chosen, the MeGraph outputs the baselines by a large margin.

---

> > > > ### Author Response · Authors · 2022-12-11
> > > > **Follow-up (Part 2/2)**
> > > >
> > > > We change the argument of GFuN so that it behaves like GAT and GatedGCN (denoted as: with GAT and GATED).
> > > >
> > > > The results of the Graph Theory Benchmark are as below:
> > > >
> > > > | Category           | Model                     | SP$_{sssd}$ |       MCC       |    Diameter     | SP$_{ss}$ |       ECC       |
> > > > | ------------------ | ------------------------- | :---------------: | :-------------: | :-------------: | :-------------: | :-------------: |
> > > > | MeGraph with GAT   | $h=1$                     |   2.990 ± 3.411   |  3.346 ± 3.228  | 44.406 ± 36.325 | 16.387 ± 13.572 | 29.039 ± 27.589 |
> > > > | MeGraph with GATED | $h=1$                     |   4.144 ± 4.181   | 0.9078 ± 0.9340 |  6.343 ± 7.152  | 13.935 ± 12.777 | 19.734 ± 19.474 |
> > > > | MeGraph with GAT   | $h=5$                     |  0.5936 ± 0.9025  |  1.706 ± 1.409  |  3.256 ± 2.956  |  1.018 ± 1.071  | 14.800 ± 17.091 |
> > > > | MeGraph with GATED | $h=5$                     |  0.8093 ± 0.9925  | 0.6597 ± 0.6012 |  2.506 ± 2.639  | 0.6690 ± 0.5457 |  7.508 ± 7.558  |
> > > > | MeGraph with GAT   | $h=5,\eta_v=0.3,\tau_c=4$ |  0.7489 ± 1.131   | 1.128 ± 0.7937  |  4.430 ± 4.329  | 0.6403 ± 0.8331 |  5.649 ± 4.496  |
> > > > | MeGraph with GATED | $h=5,\eta_v=0.3,\tau_c=4$ |  0.6017 ± 0.6217  | 0.5992 ± 0.5204 | 0.5442 ± 0.4901 | 0.3417 ± 0.1934 | 0.8593 ± 0.7122 |
> > > >
> > > > There are consistent improvements compared with the h=1 baseline no matter the core GNN used and which pooling arguments are used.
> > > >
> > > > ---
> > > > UPD (Dec 11): GNN Benchmark Results, the improvements are also consistent:
> > > > | Category           | Model |      ZINC*      |     AQSOL*      |    CIFAR10     |     MNIST      |    PATTERN     |    CLUSTER     |
> > > > | ------------------ | :---- | :-------------: | :-------------: | :------------: | :------------: | :------------: | :------------: |
> > > > | MeGraph with GAT   | $h=1$ | 0.4258 ± 0.0054 | 1.1421 ± 0.0270 | 69.890 ± 0.209 | 97.570 ± 0.168 | 78.232 ± 0.827 | 59.497 ± 0.207 |
> > > > | MeGraph with GAT   | $h=5$ | 0.3637 ± 0.0116 | 1.0767 ± 0.0105 | 69.925 ± 0.631 | 97.860 ± 0.098 | 83.798 ±0.885  | 68.93 ± 68.762 |
> > > > | MeGraph with GATED | $h=1$ | 0.3336 ± 0.0036 | 1.0766 ± 1.0556 | 64.200 ± 0.586 | 96.812 ± 0.205 | 85.391 ± 0.029 | 59.321 ± 0.290 |
> > > > | MeGraph with GATED | $h=5$ | 0.2897 ± 0.0291 | 1.0240 ± 0.0098 | 64.935 ± 0.829 | 97.29 ± 0.140  | 86.611 ± 0.041 | 67.122 ± 3.323 |

---

> > > > ### Author Response · Authors · 2022-12-13
> > > > **Please consider our most recent responses**
> > > >
> > > > Dear Reviewer M2DM,
> > > >
> > > > We sincerely thank you for your responses and suggestions. The reviewer-author discussion phase is being closed today, and we might not have further chances to hear from you. If you are satisfied with our last replies, we would be grateful if you could consider updating the score to reflect this. We understand that the review process can be demanding, and we appreciate the time and effort that you have put into providing feedback on our submission.
> > > >
> > > > Best,
> > > >
> > > > The authors

---

> ### Author Response · Authors · 2022-11-28
> **Looking forward to further feedback**
>
> Dear Reviewer M2DM,
>
> Since the author-reviewer discussion period has been launched for a few days, we appreciate if you could read our responses soon. Then, if you have any further questions and comments, we could still be able to reply before the author-reviewer discussion deadline. If our response resolves your concerns, we kindly ask you to consider raising the rating of our work. Thank you very much for your time and efforts!
>
> Best,
>
> The authors

---

### Official Review · Reviewer_F7rn · 2022-10-23

**Confidence:** 4
**Correctness:** 4
**Technical Novelty And Significance:** 3
**Empirical Novelty And Significance:** 4
**Recommendation:** 8

**Clarity, Quality, Novelty And Reproducibility:**

- Clarity: the paper is clear and well-written.
- Quality: the work is of good quality, with minor issues that should be easily addressed by the authors.
- Novelty: the work proposes several novel ideas and the references to previous work are complete.
- Reproducibility: it should be possible to implement the main method and reproduce the results from what is described in the paper. The authors mentioned that the code would be provided separately but I could not find it (I suppose OpenReview does not allow posting comments until the reviews are released). I encourage the authors to provide the anonymized code before final recommendations are submitted.

**Strength And Weaknesses:**

**Strengths**:
- The paper presents numerous novel ideas related to graph neural networks and, more specifically, graph pooling.
- The design of the proposed method is well-motivated and validated through an ablation study.
- The performance improvements are significant and, in certain cases, MeGraph outperforms the baselines by a large margin.
- The experimental section is very thorough with convincing results.

**Weaknesses**:
- What is the meaning of the following sentence?
  > In the process stage, an elementary component is a layer shaped like a mirrored E
- One of the motivations for the Mee layer is overcoming the need of having multiple GCN steps to enable communication between layers of the pyramid (the naive message-passing strategy). However, because each step in Mee diffuses information between adjacent layers of the pyramid, this issue is not really solved (it's just hidden "inside" the layer). Have I misunderstood something? Could a more efficient inter-layer communication be achieved via dense self-attention between all pyramid layers?
- The Expand function is commonly referred to as "lifting" the graph. The authors should either justify why they renamed the function or change the name to "Lift" (or something along those lines).
- While the experimental results are interesting, it is not clear if the baselines reported from other works are comparable to the proposed method (in terms of the number of parameters, number of layers, etc). Can the authors comment on this?
- Since computational cost is an issue (as commented by the authors in Section 6), the authors should report a comparison of the actual computational times of their architecture compared to the baselines.
- The color-coding of results in Tables 1, 2, and 3 is unclear since the differences between the results are small and the differences in color are barely perceptible.

**Summary Of The Paper:**

The paper presents a novel architecture for graph neural networks called MeGraph.

MeGraph builds on the Select-Reduce-Connect framework for graph pooling (Grattarola et al., 2022).
In particular, the Select function of a pooling method implicitly induces a bipartite "inter-graph" between the input and output graph. By joining all graphs in the pyramid with the corresponding inter-edges, we obtain a pyramidal "mega-graph".

By applying a specifically designed message-passing scheme (called Mee) to the mega-graphs, the authors allow for long-range communication between nodes and inter-layer communication across the pyramid.
To do so, the authors extend the SRC framework with an "Expand" function that enables backward communication from higher-level (coarse) layers to lower-level (fine) ones.

The authors also propose a novel pooling technique that extends EdgePool by allowing for arbitrary coarsening ratios.

Finally, the authors perform an in-depth experimental analysis on several relevant benchmarks, also introducing 5 new synthetic datasets to explicitly test long-range communication in GNNs.

The results show that the proposed method outperforms relevant baselines.

**Summary Of The Review:**

The work presents several novel and interesting ideas to improve the performance of graph neural networks (especially concerning long-range communication). The proposed method makes smart use of graph pooling and achieves good results in many diverse experiments.

I recommend acceptance, conditional on the authors addressing the minor concerns that I have raised above.

---

> ### Author Response · Authors · 2022-11-18
> **Respond to Reviewer F7rn**
>
> Thank you for the thorough review.
> > **1**. What is the meaning of the following sentence? ... a layer shaped like a mirrored E.
>
> **Answer**: This sentence explains the name of the Mee layer. An Mee layer (e.g., with height h=3) is shaped similar to the symbol: a mirrored $\mathbb{E}$ (horizontally flipped $\mathbb{E}$)
>
> > **2**. (About Mee layer) ... Have I misunderstood something? Could a more efficient inter-layer communication be achieved via dense self-attention between all pyramid layers?
>
> **Answer**: Thanks for this constructive suggestion. The motivation of Mee Layer is overcoming the need of stacking multiple **network layers** (i.e., applying the naive message-passing strategy **over the mega graph**) to enable communication between graphs at different heights of the pyramid.
>
> It is a good idea to apply dense self-attention among all graphs at different scales to achieve more straightforward communication. In fact, we considered some similar structures before the current MeGraph version. However, dense connections among multi-scale graphs could amplify the computational complexity issue when _dense_ graph pooling methods (as classified by Grattarola et al.) are used, explained in detail below.
>
> For a graph pooling method, let the number of nodes in the input graph be $N$, and the node pooling ratio is $\eta_v$ on average, then the number of nodes in the pooled graph is $\eta_v N$. Assume the _density_ (defined by Grattarola et al.) of the pooling method is $k$, which means each node in the input graph maps to $k$ nodes in the pool on average. Then for a $h$-height graph pyramid, the expected number of inter-graph edges between all scales is approximately $\sum_{i=1}^{h-1} \sum_{j=1}^i N \min(\eta_v^{i+j-1} N, k^j)$, which is $O(N k^h)$ when $\eta_v^h N > k^h$ (assume $\frac{1}{1-\eta_v}$ is a constant). This number could exceed the number of edges in the original graph and increase the overall time complexity, even when $k$ is a small number like 2 or 3. In contrast, the number of inter-graph edges between consecutive scales (used by the Mee layer) is $O(Nk)$.
>
> Considering the compatibility of different graph pooling methods, we finally chose the back-and-forth way with lower computational complexity to propagate information.
>
> > **3**. The Expand function is commonly referred to as "lifting" the graph.
>
> **Answer**: Thanks for pointing it out. We were unaware of this term and used "Expand" as an inverse action of "Reduce" (following the SRC framework).
>
> > **4**. It is not clear if the baselines reported from other works are comparable to the proposed method.
>
> **Answer**:
> Due to different settings, implementation details, and training tricks, the numbers directly fetched from other works are not rigorously comparable. To achieve a fair comparison, we follow the experimental protocol stated in Sec 4.4. We first try our best to reproduce the GCN result, then replace GCN with GFuN and perform a hyper-parameters tuning over that. We call this model the MeGraph (h=1) baseline, which is rigorously comparable to our h>1 version of MeGraph.
>
> Please also refer to our posted **General Message** about comparing baselines.
>
> > **5**. The authors should report a comparison of the actual computational times of their architecture compared to the baselines.
>
> **Answer**: We have reported the actual running time in Appendix D: "The practical running time of MeGraph model with height h>1 after optimization is about 2h times as the h=1 **baseline**". Below, we provide a concrete running time for one epoch on the GNN benchmark and OGBG datasets.
>
> |               | dataset    |  ZINC* | AQSOL* | CIFAR10 | MNIST  | PATTERN | CLUSTER |
> | ------------- | :--------- | :----: | :----: | :-----: | :----: | :-----: | :-----: |
> | Megraph (h=5) | time(s)    | 25.69  | 20.22  | 336.63  | 307.23 | 101.52  |  69.65  |
> | Megraph (h=1) | time(s)    |  2.41  |  1.67  |  51.74  | 38.60  |  9.21   |  6.52   |
>
> *: We use h=3 for ZINC and AQSOL.
>
> |               | dataset    | molhiv  | molbace | molbbbp | molclintox | molsider |
> | ------------- | :--------- | :-----: | :-----: | :-----: | :--------: | :------: |
> | Megraph (h=5) | time(s)    | 393.42  |  14.70  |  20.36  |   14.03    |  14.12   |
> | Megraph (h=1) | time(s)    |  22.50  |  1.43   |  1.58   |    1.26    |   1.41   |
>
> |               | dataset    | moltox21 | moltoxcast | molesol | molfreesolv | mollipo |
> | ------------- | :--------- | :------: | :--------: | :-----: | :---------: | :-----: |
> | Megraph (h=5) | time(s)    |  70.15   |   78.77    |  11.68  |    6.24     |  44.77  |
> | Megraph (h=1) | time(s)    |   5.27   |    8.17    |  0.76   |    0.41     |  2.77   |
>
>
> > **6**. The color-coding of results in Tables 1, 2, and 3 is unclear since the differences between the results are small and the differences in color are barely perceptible.
>
> Answer: Thanks for pointing it out. To make it clearer, we have emphasized the best result in bold in the revision.

---

> > ### Comment · Reviewer_F7rn · 2022-12-12
> > **Reply**
> >
> > I thank the authors for their reply. I have no further comments.

---

### Official Review · Reviewer_nn8y · 2022-10-24

**Confidence:** 3
**Correctness:** 4
**Technical Novelty And Significance:** 4
**Empirical Novelty And Significance:** 4
**Recommendation:** 8

**Clarity, Quality, Novelty And Reproducibility:**

The paper is reasonably well-written and provides considerable detail and a level of description for reproducibility. Despite that, the authors state that they will share their computer code.

**Strength And Weaknesses:**

The paper proposes a novel approach to aggregate graph neural networks. State-of-the-art results on various synthetic and natural datasets, incorporating some novel benchmarks. The authors promise that the code is going to be available. Throughout the article, particularly in the appendix, the authors describe the different aspects of and around their proposal.

There are some minor editing details, which include:
***” However, if the architecture could inference from a larger scope, e.g., constructing multi-scale graphs in a hierarchy, the shortest path is easier to be
estimated by aggregating and delivering information from multi-level scopes.” Infer?
*** convoluations
*** SEELCT
*** Is layer n in Figure 3 missing some symbols because they are white? I am talking about the nodes X_i^{n-???}. This problem also happens to the first column of nodes in Figure 4.

**Summary Of The Paper:**

The paper proposes a method to create multi-scale graphs using graph pooling. In the process, they construct a mega-graph. Their approach applies graph convolutions to intra-graph edges, while convolutions over inter-graph edges transfer information along the hierarchy. They extend graph convolutions and pooling. Their experimental results include 55 synthetic datasets and 26 real datasets. Their results outperform the chosen baseline algorithms. To share the code, they will share a link in an official comment on OpenReview. The article provides ample detail on the datasets, methods and architectures, implementation, and experimental results in the appendix.

**Summary Of The Review:**

The paper presents a novel idea that is sustained on state-of-the-art results with respect to baseline alternatives on a range of synthetic and natural datasets. On top of a detailed description, the authors offer to share their computer code.
I would invite the authors to edit the manuscript to improve their presentation.

---

> ### Author Response · Authors · 2022-11-18
> **Respond to Reviewer nn8y**
>
> Thanks for carefully reading our manuscript and pointing out these typos and minor issues. We have revised the paper accordingly.
>
> Thanks for acknowledging the contribution of our work. Please check our **General Message** and let us know if there are any other questions.

---

### Official Review · Reviewer_AA1h · 2022-10-29

**Confidence:** 3
**Correctness:** 2
**Technical Novelty And Significance:** 2
**Empirical Novelty And Significance:** 1
**Recommendation:** 3

**Clarity, Quality, Novelty And Reproducibility:**

The presentation could be improved and the paper could be better organized. The current version of this manuscript is not very easy to follow. The proposed methods sound novel but it is a mostly incremental improvement over a combination of previous methods. Code is provided.

**Strength And Weaknesses:**

Cons:

1. The algorithm studied in this paper is quite complicated, it has many technical details and is very hard to digest, and requires massive ablation study. By looking at their ablation study results, I am not confident about which part of the proposed modules could really boost the performance. Although the authors provide massive experiment results in the appendix, I would suggest the authors summarize the main take-home messages in bullet points (e.g., by comparing results A with B we know using module C could roughly improve D percent performance.) Keep in mind to make the religious argument and avoid over-claiming the contributions when conducting an ablation study.

2. The multi-scale graph learning issue in this paper could be very related to jumping knowledge [1] and its follow-up works. According to my understanding, the biggest difference is this paper allows information exchange across multi-scale graphs and a different type of graph convolution operator. I think authors might also need to compare with those types of works, instead of just comparing to vanilla GCN and GIN, to help readers better understand the advantages of this work.

[1] Representation Learning on Graphs with Jumping Knowledge Networks  https://arxiv.org/abs/1806.03536

3. The authors mention many related methods, e.g., Graph U-Net and various pooing methods. These methods are expected to compare with the experiment to understand how much MeGraph gains from these baselines.

4. If possible, please also compare with OGB's node classification baseline methods on node classification datasets such as OGB-Arxiv, and OGB-Products. The algorithm developed for OGB node classification tasks is usually more advanced and more challenging.

5. The impact and contributions of this work might not be enough for ICLR.
Although some new techniques are introduced in this paper, these methods are not fundamental enough, are mostly at the ML engineering aspect, and could have very limited impact in practice.


**Summary Of The Paper:**

This paper studies graph representation learning and propose MeGraph that could learn multi-scale graph representation. MeGraph enables information exchange across multi-scale graphs, which distinguishes MeGraph from many recent hierarchical graph neural networks.
To achieve this goal, MeGraph first uses graph pooling to create multi-scale graphs, then convolve the intra-graph edges and inter-graph edges separately, and finally adopts graph full network (GFuN) and stridden edge contraction pooling (S-EdgePool) for feature extraction.

**Summary Of The Review:**

This paper is an empirical paper working on graph representation learning. The algorithm studied in this paper is quite complicated, it has many technical details and is very hard to digest. By looking at their ablation study results, I am not convinced how MeGraph could improve the model performance. There are baselines and related methods that need to be compared with given this method is also quite complicated and related to many other existing strategies. The impact and contribution of this paper might not enough to reach the bar of ICLR.

---

> ### Author Response · Authors · 2022-11-18
> **Respond to Reviewer AA1h**
>
> According to your summary of our work, there might exist some misunderstanding of the working principle of MeGraph. We hope our answers below cloud help eliminate the confusion.
>
> > **Summary in the review**. To achieve this goal, MeGraph first uses graph pooling to create multi-scale graphs, then convolve the intra-graph edges and inter-graph edges separately, and finally adopts graph full network (GFuN) and stridden edge contraction pooling (S-EdgePool) for feature extraction.
>
> **Clarification**:
> The working principle of MeGraph does not follow the "first(pooling)-then(convolution)-finally(GFuN and S-EdgePool)" order. Graph poolings and convolutions are not trained separately. The structure of the mega graph (i.e., multi-scale graphs and connections between the successive scales) is not fixed but evolved along with the training of graph pooling and convolutions. We noticed that our presentation in the introduction used some transitional words, which led to this confusion. We have revised the contexts accordingly.
>
> > **1a**. The algorithm studied in this paper is quite complicated, it has many technical details and is very hard to digest.
>
> **Answer**:
> Briefly, the overall algorithm is to build the mega graph using differentiable graph pooling methods and apply GNNs over the mega graph simultaneously (summarized in the first two sentences of Section 3). We introduced how to obtain multi-scale graphs using graph poolings and how to build the mega graph using pooling results in Sec 3.1.
>
> MeGraph utilizes Mee layers with bidirectional pathways to achieve efficient multi-scale information exchange (Figures 3 and 4 and their captions provide the general scope of MeGraph).
>
> Please refer to Sections 2 and 3 for a more detailed description of the method. The first sentence of each section could provide a rough picture of MeGraph.
>
> If there still exist difficulties in digesting the paper, please let us know the specific points that need to be clarified, and we will be happy to explain them in detail and improve our presentation.
>
> > **1b**. Requires massive ablation study.
>
> **Answer**:
> We have conducted a number of ablation studies concerning the following three perspectives:
> 1. Investigating the influence of the key hyper-parameters: Section 4.2 provides grid search results for the height $h$ and the number of layers $n$.
> 2. Investigating the structure of MeGraph: we compared various MeGraph variants by varying its shape (e.g., we provided h=1, n=1 and U-Shaped variants), which are also regarded as reliable baselines (in Section 4.1). We perform comprehensive experiments on both the graph theory benchmark and real-world datasets.
> 3. Replacing the specific components: we compared the S-EdgePool and EdgePool (different pooling methods, shown in Table 1), X-Conv and X-Pool (different inter-graph update methods, explained in the revised version of Mee Layer in Section 3.2, shown in Table 1).
>
> There exist more candidate structures and components in the literature, but it is impractical for us to enumerate all possible choices and combinations. If possible, please give us some specific suggestions if the current ablation studies are still inadequate and unclear.
>
> > **1c**. I would suggest the authors summarize the main take-home messages in bullet points.
>
> **Answer**: Thanks for your suggestion. We have refined the presentation of our ablation study results in Section 4.3 accordingly.
>
> > **2**. The multi-scale graph learning issue in this paper could be very related to jumping knowledge [1] and its follow-up works.
>
> **Answer**: Thanks for pointing out the jumping knowledge network (JK). The jumping knowledge network mainly targets aggregating information within different neighborhood ranges to enable better structure-aware representation. This differs from our method because different neighborhood ranges (varying by layer indexes) are not equivalent to multi-scale graphs (varying by heights). Moreover, the way to update information in these two methods differs (central for JK and pathway for MeGraph).
>
> Indeed, MeGraph can adopt the jumping knowledge layer aggregation when stacking layers, so JK considers an orthogonal direction compared to MeGraph.
> We will create a new section about more related work in the Appendix to discuss the relationship between JK and MeGraph.

---

> > ### Author Response · Authors · 2022-11-18
> > **Respond to Reviewer AA1h Cont'd**
> >
> > > **3**. These methods are expected to compare with the experiment.
> >
> > **Answer**: In Sec 4.1 (Baselines), we have compared a U-Shaped variant under the MeGraph architecture, which shares the same graph pooling method and GN block as MeGraph for a fair comparison. The results are shown in Table 1 ([n=9, U-Shaped] 6.656 vs. [n=5] 2.337 MSE). The original Graph U-Nets paper also reports performance on some TU datasets, while the numbers (inferior to ours) are not directly comparable due to different settings. We further run such a U-shaped variant and our reproduced Graph U-Nets (with the same pooling and convs as MeGraph) on GNN benchmark and OGBG datasets. We present these results in the follow-up comment of the **General Message**.
> >
> > > **4**. If possible, please also compare with OGB's node classification baseline methods on node classification datasets such as OGB-Arxiv, and OGB-Products.
> >
> > **Answer**: We have already presented massive experimental results over the Graph Theory Benchmark (ours), GNN benchmarks, OGB-G datasets, and TU datasets. Our method is more effective on tasks requiring long-range relationships (as shown in Table 1).
> > The _OGBN-Arxiv_ is a node classification task on a citation network in which closely related nodes have already been linked by edges (citations). Therefore, long-range relationships are probably optional within this task. The _OGBN-Products_ dataset shares a similar characteristic. In fact, the aforementioned properties of the two datasets have already been discussed in the reference (Uri Alon and Eran Yahav) provided by Reviewer M2DM.
> >
> > Nonetheless, per you request, we have supplemented experiments on the OGBN-Arxiv and OGBN-Products datasets.
> > We split the whole graph into ten parts in OGBN-Arxiv and split the entire graph into 10000 parts in OGBN-Products following the Cluster-GCN (Chiang, Wei-Lin, et al.).
> > The results are provided in the following table. As expected, the performance of MeGraph shows slight superiority compared to the baselines, while it is still among the most competitive candidates.
> >
> > | Method   |     GCN      | GFN          | Megraph (h=1) |     Megraph      |
> > | :------- | :----------: | :----------- | :-----------: | :--------------: |
> > | arxiv    | 71.75 ± 0.26 | 72.13 ± 0.21 | 72.37 ± 0.27  | **72.57 ± 0.18** |
> > | products | 75.36 ± 0.43 | 75.51 ± 0.25 | 77.81 ± 0.45  | **78.32 ± 0.30** |
> >
> > Uri Alon and Eran Yahav. On the bottleneck of graph neural networks and its practical implications. In International Conference on Learning Representations, 2021.
> >
> > Chiang, Wei-Lin, et al. "Cluster-gcn: An efficient algorithm for training deep and large graph convolutional networks." Proceedings of the 25th ACM SIGKDD international conference on knowledge discovery & data mining. 2019.
> >
> > > **5a**. These methods are not fundamental enough, are mostly at the ML engineering aspect, and could have very limited impact in practice.
> >
> > > **5b**. The proposed methods sound novel but it is a mostly incremental improvement over a combination of previous methods.
> >
> > > **5c**. This paper is an empirical paper working on graph representation learning.
> >
> > **Answer**: Would you please provide us a few measurable points or improvement suggestions for MeGraph to become sufficiently fundamental so that we can improve the work accordingly?
> >
> > Currently, we could not agree that MeGraph is mostly at the ML engineering aspect and mostly incremental improvement over a combination of previous methods. Please refer to the posted **General Message** about novelty. We think MeGraph is novel, considering the construction of the mega graph, the operations we have developed over the mega graph, and its superior performance in a wide range of tasks.
> >
> > In addition, MeGraph is indeed implemented with careful coding efforts. For example, we used the disjoint-set data structure to efficiently implement the S-EdgePool operator (as mentioned in Section 3.3). We carefully implemented this algorithm to support batch operation. We also did constant optimization and used Taichi-Lang to further speed up the computation (stated in Appendix D). These engineering efforts speed up the running time of MeGraph by a large margin (about 5x faster). So, we think engineering efforts should not be less credited. There are also many famous neural networks with heavy efforts on the engineering part, such as the Swin Transformer (Z. Liu, Y. Cao, et al.).
> >
> > Z. Liu, Y. Lin, Y. Cao, H. Hu, ... & B. Guo. Swin transformer: Hierarchical vision transformer using shifted windows. In Proceedings of the IEEE/CVF International Conference on Computer Vision, 2021.

---

> ### Author Response · Authors · 2022-11-28
> **Looking forward to further feedback**
>
> Dear Reviewer AA1h,
>
> Since the author-reviewer discussion period has been launched for a few days, we appreciate if you could read our responses soon. Then, if you have any further questions and comments, we could still be able to reply before the author-reviewer discussion deadline. If our response resolves your concerns, we kindly ask you to consider raising the rating of our work. Thank you very much for your time and efforts!
>
> Best,
>
> The authors

---

> ### Author Response · Authors · 2022-12-11
> **Looking forward to further feedback**
>
> Dear Reviewer AA1h,
>
> The author-reviewer discussion period is approaching the end, and we would appreciate if you could let us know whether our responses are satisfactory, and whether there are any further questions or comments. If our answers resolve your concerns, we kindly ask you to consider raising the rating of our work. Thank you very much again for your time and efforts!
>
> Best,
>
> The authors

---

> > ### Comment · Reviewer_AA1h · 2022-12-11
> > **To authors**
> >
> > I will remain by score because my concerns are not addressed. Please refer to the following reasons :
> >
> > I am not confident whether the performance is really coming from the proposed methods (e.g., using differentiable graph pooling to generate multi-scale graphs, connecting multi-scale graphs into meta-graphs, graph full network, stride edge contracting padding) and which part of the proposal really contributes the most to the model performance. The authors try to address my concern by pointing to the ablation studies they made, including hyper-parameter selection, different model architecture/backbone, and different pooling. Unfortunately, I cannot conclude from their results that “we solve the XXX issue because using differentiable graph pooling to generate multi-scale graphs, connecting multi-scale graphs into meta-graphs, graph full network, stride edge contracting padding”. This is just my own opinion but I understand it is not easy to justify especially since the algorithm is advanced with many novel modules and techniques.
> >
> > Results on the challenging dataset are not good. For example, on the OGB-Arxiv (https://ogb.stanford.edu/docs/leader_nodeprop/#ogbn-arxiv) the best results are 79.66% accuracy score and even MLP with post-processing (C&S) can achieve 73.12%, but the one reported in the rebuttal in 72.57%; on the OGB-Products (https://ogb.stanford.edu/docs/leader_nodeprop/#ogbn-products) the best results are 90.14% accuracy score and vanilla GraphSAGE can achieve 78.29% without using any tricks as proposed in this paper, but the one reported in the rebuttal in 78.32%; The authors are selecting relatively weak baseline models on the benchmark in rebuttal. Please refer to the link above for the public benchmark score.
> >
> > In terms of the importance of this paper, I think the main take-home message is a new multi-scale graph neural network modeling algorithm, but I am not sure how future studies could benefit from this work (after reading authors feedbacks) because I am not confident about whether the proposed method could truly work in practice give the above two points. However, this is just my own opinion and I wouldn’t be sad if the AC decide to accept this paper.
> >
> > Reviewer

---

> > > ### Author Response · Authors · 2022-12-12
> > > **Follow-up**
> > >
> > > Thanks for your response. we provide a point-by-point response below.
> > >
> > > > I am not confident whether the performance is really coming from the proposed methods (e.g., using differentiable graph pooling to generate multi-scale graphs, connecting multi-scale graphs into meta-graphs, graph full network, stride edge contracting padding) and which part of the proposal really contributes the most to the model performance.
> > >
> > > > I cannot conclude from their results ...
> > >
> > > 1.  We did not claim GFuN as one of our contributions. We kept the GFuN the same core GN block when comparing the MeGraph model with the h=1, the n=1, and Graph UNets baselines (stated in Sec 4.1).
> > > 2. Please refer to Figure 5. In Sec 4.2, we mentioned that "When we set h = 1, the MeGraph model reduces to a normal GNN over the original graph" and "We can observe clear gaps among curves of h = 1, h = 2 and h ≥ 3 for all values of n". These results **clearly** indicate the effectiveness of the MeGraph architecture in the TreeCycle and TreeGrid tasks, and the results are consistent no matter how many layers are used. Therefore, the performance gain of MeGraph benefits from using the differentiable graph pooling and the mega-graph structure (instead of 'meta-graph' as in your comments).
> > > 3. The same conclusion can be drawn by comparing the performance of the MeGraph (h=1) baseline and the MeGraph model in Tables 1, 2, and 3.
> > > 4. When comparing the MeGraph model with the n=1 and Graph UNets baselines (in Tables 1, 2, and 3 and the tables in the posted General Message), we can conclude the advantage of enabling repeated information exchange across multi-scale graphs.
> > > 5. As for S-EdgePool, please see Table 1. While keeping $h=5$ and $n=5$, we vary the arguments $\tau_c$ and $\eta_v$ of S-EdgePool where the $\tau_c=2$ represents the original EdgePool. The best-performing S-EdgePool (with an average error of 0.624) outperforms the original EdgePool (with an average error of 2.337) by a large margin.
> > >
> > > > Results on the challenging dataset are not good.
> > >
> > > > The authors are selecting relatively weak baseline models on the benchmark in rebuttal.
> > >
> > > We cannot agree with the comments for the following reasons.
> > > 1. The results you mentioned are NOT directly comparable. For OGBN-arxiv, MLP with post-processing (C&S) could achieve 73.12%. However, we didn't adopt C&S when running our results (which is orthogonal and can be used in our methods). Also note that we use Cluster-GCN to sample sub-graphs, which is different from how MLP+C&S is trained. Therefore, they are not directly comparable due to different settings. For OGBN-product, GraphSAGE could achieve 78.29%. However, it uses FullNeighborSampler (https://github.com/dmlc/dgl/blob/master/examples/pytorch/ogb/ogbn-products/graphsage/main.py#L67), which is different from the Cluster-GCN sampler and should not be directly compared either.
> > > 2. We would like to emphasize that our focus is **verifying the effectiveness** of the novel MeGraph architecture in **most comparable settings**, rather than achieving SOTA on OGBN datasets. Therefore, it is not necessary to achieve a good rank over the OGB leaderboard using tricks and tuning lots of hyper-parameters. Moreover, we have explained in the rebuttal that "our method is more effective on tasks requiring long-range relationships" and "the OGBN-Arxiv is a node classification task on a citation network in which closely related nodes have already been linked by edges" (as already discussed by Uri Alon and Eran Yahav). The motivation behind MeGraph is not to improve the GNN results of datasets that long-range relationships are not that important (like OGBN-arxiv and OGBN-product).
> > > 3. We think a comprehensive evaluation of MeGraph should not be focused only on OGBN datasets. The proposed Graph Theory Benchmark is specifically designed to verify the effectiveness of MeGraph in identifying long-range relationships.
> > >
> > > > but I am not sure how future studies could benefit from this work ... because ...
> > >
> > > Please refer to the reponse above and also the future work section (Sec. 6).
> > > 1. MeGraph makes it possible to design "new graph pooling methods to yield (inter-graph) edge features in addition to node features," where inter-graph edge features could also get gradient through backpropagation.
> > > 2. It is interesting to explore MeGraph along with adaptive computational steps for better generalization performance.
> > > 3. It is also possible to apply some (computational) expressive models like Transformers and Neural Logic Machines over the pooled small-sized graphs to further increase the expressiveness of the model.

---

> > > ### Author Response · Authors · 2022-12-13
> > > **Please consider our most recent responses**
> > >
> > > Dear Reviewer AA1h,
> > >
> > > We sincerely thank you for your responses and suggestions. The reviewer-author discussion phase is being closed today, and we might not have further chances to hear from you. If you are satisfied with our last replies, we would be grateful if you could consider updating the score to reflect this. We understand that the review process can be demanding, and we appreciate the time and effort that you have put into providing feedback on our submission.
> > >
> > > Best,
> > >
> > > The authors

---

### Author Response · Authors · 2022-11-18
**General Message**

We thank all reviewers for your invaluable reviews. We provide point-by-point responses below by commenting on each of your reviews. We have also revised the manuscript accordingly and annotated the differences in blue color. We provide an anonymous code link in an independent comment (only visible to reviewers). We will make the code public after the open-review period.
## Novelty
We would like to re-clarify our main contributions.
1. **We provide a novel perspective of a mega graph for multi-scale graph representation learning problems**. Conditioning on the graph pooling results, we explicitly connect multi-scale graphs into a mega graph according to how the nodes are pooled together (detailed in Sec 3.1). This enables us to apply graph convolutions over the connections between different scales instead of using pooling and unpooling operations to update representations (we have studied the difference of applying graph convolutions and simple pooling and unpooling operations in Sec 4.3). Moreover, we would like to clarify that the structure of the mega graph (i.e., the structure of multi-scale graphs and connections between the successive scales) is not fixed but evolved along with the training of graph pooling and convolutions.
2. **We utilize the hierarchical nature of the mega graph and design a novel Mee layer that achieves efficient cross-scale update**  (detailed in Sec 3.2). We would like to emphasize the novelty within the two pathways of the Mee layer (Figure 4 illustrates the detailed architecture), which were not used for multi-scale graph learning before. Without the pathways, it could take $h-1$ layers for the information of the original graph to pass to the the smallest graph, which is unacceptable when $h$ goes large.
3. **Using a novel cluster-finding algorithm, we propose S-EdgePool that addresses the issue of the original EdgePool where the pooling ratio is fixed** (detailed in Sec 3.3). In the ablation study between S-EdgePool and EdgePool in Sec 4.3, we show the importance of having a flexible pooling ratio. We also provided an efficient implementation of S-EdgePool to speed up the pooling operation.
4. We create a graph theory benchmark consisting of 55 datasets to test models' ability to handle long-range relationships. **In this benchmark, MeGraph demonstrates dominated performance compared with the baselines**.

In summary, MeGraph is NOT a combination of existing GNN modules (including graph convs and graph poolings); instead, it provides a novel framework for multi-scale graph representation learning, which expertises in tasks that require considering long-range relationships. These modules are NOT trivially composed together where the structure of the mega graph keeps evolving along with the training of the graph pooling and convolution operations.

We will make our contributions clearer in the revision.

## About comparing with other methods
MeGraph adopts common GNN modules (as summarized in (You et al.)) as components, including graph convs (GN block), graph poolings, graph encoders (including positional encoding and graph rewiring), and graph decoders (readout functions). These components are free to be chosen, and there is no need to stick to specific choices. So, **MeGraph is orthogonal to advances in these modules** and could benefit from modular improvements.

Another empirical fact is that the exact experimental settings (including the choices of the GNN modules) for these approaches differ from each other. It is not trivial to provide a unified platform for exhaustive comparison.

In this work, we focus on **comparison of structural variations while keeping the modules invariant**. We elaborate on various cases (detailed in Sec 4.1) by varying the number of scales (height), the number of layers, and the architecture connections (including the U-shaped variant). These baselines are rigorously comparable as they are implemented in the same codebase under the same setting.

As suggested by reviewers, we have re-implemented Graph U-Net within our codebase, using the same GN block (GFuN) and graph pooling (S-EdgePool) methods as MeGraph. We report the results in a follow-up comment under the most comparable settings.

You, J., Ying, Z., & Leskovec, J. (2020). Design space for graph neural networks. Advances in Neural Information Processing Systems, 33, 17009-17021.

---

> ### Author Response · Authors · 2022-11-19
> **U-shaped Variants and Re-implemented Graph U-Nets Results**
>
> We have re-implemented Graph U-Net within our codebase, using the same GN block (GFuN) and graph pooling (S-EdgePool) methods as MeGraph. We present the results of U-Net and U-shaped variants (with X-Pool and X-Conv variants, see also the revised part of Sec 3.2) on the GNN benchmark and OGBG datasets below (we copy the MeGraph results here for convenience, more results can be found in Tables [1-3, 7-8]).
>
> Note that the U-Shaped variants and U-Net have a fixed 2:1 ratio between the number of layers and the height (following the original Graph U-Nets (Gao & Ji, 2019) ). We make them the same height as MeGraph, therefore they actually have more layers (10 vs. 6 in the Graph Theory Benchmark, 10 vs. 4 in the GNN benchmark and 10 vs. 5 in OGBG datasets). MeGraph outperformed the U-shaped variants and reproduced Graph U-Net in most datasets (even when they have more layers).
> ## Graph Theory Benchmark Results
> | Category | Model Parameters                                        | SP$_{sssd}$ |    MCC     |  Diameter  | SP$_{ss}$|    ECC     |
> | -------- | ------------------------------------------------------------- | :---------------: | :--------: | :--------: | :-------------: | :--------: |
> | Megraph  | $h=5,n=5,\eta_v=0.3,\tau_c=4$                                 |    **0.5837**  | **0.5646** | **0.5173** |   **0.4751**    | **0.9249** |
> | U-Net    | $h=5,n=9,\eta_v=0.3,\tau_c=4$                                 |    1.115       |   0.8252   |   2.038    |      1.106      |   2.568    |
> ## GNN Benchmark Results
>
> | dataset        |        ZINC         |        AQSOL        |      CIFAR10       |       MNIST        |      PATTERN       |      CLUSTER       |
> | :------------- | :-----------------: | :-----------------: | :----------------: | :----------------: | :----------------: | :----------------: |
> | Megraph        | **0.2597 ± 0.0053** | **1.0017 ± 0.0210** | **69.925 ± 0.631** | **97.860 ± 0.098** | **86.507 ± 0.067** | **68.603 ± 0.101** |
> | U-Shape (pool) |   0.4117 ± 0.0101   |   1.0806 ± 0.0169   |  51.400 ± 23.910   |  32.562 ± 36.741   |   78.825 ± 5.905   |   34.348 ± 0.959   |
> | U-Shape (conv) |   0.3628 ± 0.0123   |   1.0589 ± 0.020    |  23.950 ± 24.162   |  32.590 ± 36.789   |   85.296 ± 0.749   |   47.20 ± 5.860    |
> | U-Net          |   0.3320 ± 0.0103   |   1.0629 ± 0.0182   |  68.567 ±  0.339    |  97.130 ± 0.227    |   86.257 ± 0.078   |   50.371 ± 0.243   |
> ## OGBG Datasets Results
>
> | dataset        |      molhiv      |     molbace      |     molbbbp      |    molclintox    |     molsider     |
> | :------------- | :--------------: | :--------------: | :--------------: | :--------------: | :--------------: |
> | Megraph        |   77.20 ± 0.88   |   78.52 ± 2.51   |   69.57 ± 2.33   | **92.04 ± 2.19** |   59.01 ± 1.45   |
> | U-Shape (pool) |   77.19 ± 1.25   |   74.43 ± 2.14   |   68.11 ± 0.85   |   87.12 ± 1.18   | **59.53 ± 0.60** |
> | U-Shape (conv) |   74.33 ± 1.83   |   72.32 ± 2.34   |   66.48 ± 0.82   |   78.24 ± 1.49   |   55.10 ± 0.43   |
> | U-Net          | **79.48 ± 1.06** | **81.09 ± 1.66** | **71.10 ± 0.52** |   91.67 ± 1.69   |   59.38 ± 0.63   |
>
> | dataset        |     moltox21     |    moltoxcast    |      molesol      |    molfreesolv    |      mollipo      |
> | :------------- | :--------------: | :--------------: | :---------------: | :---------------: | :---------------: |
> | Megraph        | **78.11 ± 0.47** | **67.67 ± 0.53** | **0.886 ± 0.024** | **1.876 ± 0.058** |   0.726 ± 0.006   |
> | U-Shape (pool) |   73.92 ± 0.54   |   61.62 ± 0.33   |   1.628 ± 0.083   |   3.598 ± 0.105   |   1.069 ± 0.012   |
> | U-Shape (conv) |   75.44 ± 0.35   |   61.51 ± 0.53   |   1.463 ± 0.50    |   3.696 ± 0.108   |   1.062 ± 0.023   |
> | U-Net          |   77.85 ± 0.81   |   66.49 ± 0.45   |   1.002 ± 0.036   |   1.885 ± 0.069   | **0.716 ± 0.014** |

---

### Decision · Program_Chairs · 2023-01-20

**Decision:**

Reject

**Justification For Why Not Higher Score:**

Reviewers raised concerns such as insufficient  ablation study results given complex model, differentiating with  jumping knowledge networks, need of having multiple GCN steps, computational cost, needing theoretical analysis of the expressive power of the proposed method, limited impact and contribution. The authors addressed some concerns from the reviewers. However, in author-reviewer exchanges, two of the reviewers are still not convinced regarding multiple issues they have raised. Overall, there still seems to be a lack of enough enthusiasm among the reviewers after the author responses.

**Justification For Why Not Lower Score:**

N/A

**Metareview: Summary, Strengths And Weaknesses:**

The paper proposes a mega graph structure and introduces it to a GNN named MeGraph, which could learn multi-scale graph representation. MeGraph enables information exchange across multi-scale graphs, which distinguishes MeGraph from many recent hierarchical graph neural networks. To achieve this goal, MeGraph first uses graph pooling to create multi-scale graphs. Convolutions over inter-graph edges transfer information along the hierarchy. They then convolve the intra-graph edges and inter-graph edges separately, and finally adopts graph full network (GFuN) and stridden edge contraction pooling (S-EdgePool) for feature extraction. The authors perform an in-depth experimental analysis on several relevant benchmarks, also introducing 5 new synthetic datasets to explicitly test long-range communication in GNNs. Experimental results are provided to demonstrate the effectiveness of MeGraph. Reviewers raised concerns such as insufficient  ablation study results given complex model, differentiating with  jumping knowledge networks, need of having multiple GCN steps, computational cost, needing theoretical analysis of the expressive power of the proposed method, limited impact and contribution. The authors addressed some concerns from the reviewers. However, in author-reviewer exchanges, two of the reviewers are still not convinced regarding multiple issues they have raised. Overall, there still seems to be a lack of enough enthusiasm among the reviewers after the author responses.